# Single cell profiling of circulating autoreactive CD4 T cells from patients with autoimmune liver diseases suggests tissue imprinting

Anaïs Cardon [1,17], Thomas Guinebretière [1,17], Chuang Dong[2], Laurine Gil[2], Sakina Ado[2], Pierre-jean Gavlovsky [1], Martin Braud[1], Richard Danger[1], Christoph Schultheiß [3], Aurélie Doméné[4], Perrine Paul-Gilloteaux [4], Caroline Chevalier[5], Laura Bernier[1], Jean-Paul Judor[1], Cynthia Fourgeux [1], Astrid Imbert[6], Marion Khaldi[6,7], Edouard Bardou-Jacquet[8], Laure Elkrief[9], Adrien Lannes[10], Christine Silvain[11], Matthieu Schnee[12], Florence Tanne[13], Fabienne Vavasseur[5], Lucas Brusselle[1], Sophie Brouard[1], William W. Kwok [14], Jean-François Mosnier[1,15], Ansgar W. Lohse[16], Jeremie Poschmann [1], Mascha Binder[3], Jérôme Gournay[1,6,7], Sophie Conchon[1,18] ✉, Pierre Milpied [2,18] ✉ & Amédée Renand [1,18] ✉

Autoimmune liver diseases (AILD) involve dysregulated CD4 T cell responses against liver self-antigens, but how these autoreactive T cells relate to liver tissue pathology remains unclear. Here we perform single-cell transcriptomic and T cell receptor analyses of circulating, self-antigen-specific CD4 T cells from patients with AILD and identify a subset of liver-autoreactive CD4 T cells with a distinct B-helper transcriptional profile characterized by PD-1, TIGIT and HLA-DR expression. These cells share clonal relationships with expanded intrahepatic T cells and exhibit transcriptional signatures overlapping with tissue-resident T cells in chronically inflamed environments. Using a mouse model, we demonstrate that, following antigen recognition in the liver, CD4 T cells acquire an exhausted phenotype, play a crucial role in liver damage, and are controlled by immune checkpoint pathways. Our findings thus suggest that circulating autoreactive CD4 T cells in AILD are imprinted by chronic antigen exposure to promote liver inflammation, thereby serving as a potential target for developing biomarkers and therapies for AILD.

Autoimmune liver diseases (AILD) are rare immune-mediated chronic inflammatory diseases. The three main AILD are autoimmune hepatitis (AIH), primary biliary cholangitis (PBC), and primary sclerosing cholangitis (PSC), which are characterized by the destruction of the liver parenchyma, the intrahepatic biliary epithelial cells of small bile ducts or the intra- or extra-hepatic biliary epithelial cells of the large bile ducts, respectively. AIH, PBC, and PSC are characterized by immune infiltrates in the liver[1–7]. Autoantibodies are detectable in the serum of patients, especially for AIH and PBC. PSC is a more complex disease often associated with ulcerative colitis[8]. Although all three AILD are classified as autoimmune diseases, the degree of intensity of the autoreactive response appears to be variable, and the link with the

---

autoimmune T-cell signature is unknown. In our study, we focused primarily on AIH and PBC diseases.

In AILD, the strong association with the HLA locus, the presence of autoantibodies in serum and of T cells in damaged tissue reflect the complete adaptive immune response against self-antigens, with a central role for CD4 T cells with their helper function[9–12]. The recent demonstration of the efficacy of rituximab (anti-CD20) treatment in AIH confirms that B cells contribute to the pathogenesis of AILD[13]. However, it has been difficult to establish in-depth the immune profile of autoreactive CD4 T cells, as they are rare in the blood and not all self-antigens are known. During AILD, some target self-antigens have been described. Anti-SLA antibodies target the Sepsecs self-antigen, anti-LKM1 antibodies target the CYP2D6 self-antigen, anti-LC1 antibodies target the FTCD self-antigen and the anti-mitochondrial M2 (anti-M2) antibodies target the PDCE2 self-antigen. T cell reactivity against Sepsecs, CYP2D6, FTCD, and PDCE2 is detectable in the blood of AILD patients with a preferential Th1 cytokine profile (IFNγ)[14–17]. In a mouse model, immunization against CYP2D6 or FTCD induces liver autoimmunity[18–21]. Those studies reinforce the idea of a complete adaptive immune response against self-antigens in AILD with a central role of CD4 T cells. Recently, we were able to characterize the single cell transcriptomic profile of Sepsecs-specific CD4 T cells in the blood of AIH patients with anti-SLA antibodies[22]. Compared to Candida Albicans-specific CD4 T cells, Sepsecs-specific CD4 T cells were PD-1$^+$ CXCR5$^-$ cells and expressed high level of the B-helper gene, *IL21*, as well as the *IFNG* gene and the immunoregulatory genes *TIGIT* and *CTLA-4*, among others. In the blood of AIH patients, we found an increase in IL21- and IFNγ-producing PD-1$^+$ CXCR5$^-$ CD4 T cells, independent of the presence of specific autoantibodies (e.g. anti-SLA)[22]. The signature of these autoreactive CD4 T cells in the blood presented high similarity with the peripheral T helper cells (T$_{PH}$) involved in the local adaptive immune response (B-helper function) in the synovium of patients with rheumatoid arthritis and in other tissues during autoimmunity[23–32]. However, it was still unknown whether the T$_{PH}$ immune signature was a common feature of other liver-self-antigen-specific CD4 T cells.

The similarity between circulating Sepsecs-specific CD4 T cells and T$_{PH}$ in the tissue raised the question whether circulating auto-reactive CD4 T cells are directly linked to the reactivity in the tissue. Although T$_{PH}$ cells can also be found in smaller numbers in the blood, the site of their differentiation is unknown[22,23,33]. In liver biopsies from AIH and PBC patients, Sepsecs- and PDCE2-reactive T cells are detectable but poorly characterized; they produce TNF and/or IFNγ[34–37]. Characterizing the immune signature of T cells in tissues is a timely topic, but the dynamic link between tissues and circulation is less well understood, particularly for CD4 T cells. Liver biopsies are small and rare, making it difficult to characterize liver-autoreactive CD4 T cells during AILD in the tissue. Thus, the capacity to detect in the blood autoreactive cells directly linked to tissue pathogenesis is an interesting perspective.

Circulating Sepsecs-specific CD4 T cells and other circulating self-antigen-specific CD4 T cells have an exhausted phenotype (expressing PD-1, TIGIT and CTLA-4) which is a characteristic of T$_{PH}$ cells in damaged tissues and of tissue-resident memory (T$_{RM}$)-like CD4 T cells in the tumor environment and other tissues[22,24,30,38–44]. This phenotype may reflect chronic activation and/or residency, although the distinction between these two signatures is difficult[42,43,45–48]. This exhausted phenotype could suggest a tissue imprinting of T$_{PH}$ and Sepsecs-specific CD4 T cells found in the peripheral circulation after immune response in the tissue. We thus hypothesized that some circulating autoreactive CD4 T cells derive from an exacerbated clonal expansion in the tissue during active autoimmunity.

Here we use integrative single-cell immuno-transcriptomics to characterize three distinct liver-self-antigen-specific CD4 T cell

reactivities and expanded intrahepatic CD4 T cell clonotypes in the blood of AILD patients. These cells have an exhausted phenotype and transcriptional signature that overlap with CD4 T cells in damaged tissues during autoimmunity. In a mouse model designed to study the initiation of an immune response against an antigen expressed in the liver, we observe alterations of the transcriptional profile of intrahepatic antigen-specific CD4 T cells after local antigen reactivity, which is controlled by the immune checkpoint molecules PD-1 and CTLA-4. These results demonstrate how circulating liver-self-antigen-specific CD4 T cells are linked to tissue auto-reactivity and how the hepatic environment imprints their immune signature, with an important clinical perspective for monitoring and targeting liver auto-reactivity in the blood of patients.

## Results

### Self-antigen-specific CD4 T cell detection in AILD patients with auto-antibodies

We have previously shown that Sepsecs (SLA)-specific CD4 T cells are detectable in the blood of patients with autoimmune hepatitis (AIH) whose serum contains anti-SLA antibodies[22]. In the present study, we investigated whether we could extend those findings to autoreactive CD4 T cells specific for other antigens frequently targeted by auto-antibodies in AILD. We studied the reactivity of CD4 T cells against Sepsecs, CYP2D6 and PDCE2 in AILD (AIH, PBC and overlap AIH/PBC) patients with or without anti-SLA (SLA$^+$) or anti-LKM1 (LKM1$^+$) or anti-M2 (M2$^+$) autoantibodies. As an experimental control, we also studied the reactivity of CD4 T cells against MP65 (Candida Albicans; C.ALB), MP1 (Influenza; H1N1) and SPIKE (SARS-CoV-2) in the AILD patients (Supplementary Data 1). As previously described[22,38,49–55], detection of antigen-specific CD4 T cells was achieved after four hours of peptide stimulation in vitro. CD4 T cells that specifically recognize antigen-derived peptides upregulate CD40 ligand (CD154) and can be detected by flow cytometry (Fig. 1A and B and Supplementary Fig. 1). These reactive CD4 T cells were memory cells (CD45RA$^-$; mCD4 T cells) expressing high levels of PD-1 but not CXCR5 (Fig. 1B and C and Supplementary Fig. 1). The frequency of Sepsecs-, CYP2D6- and PDCE2-specific PD-1$^+$CXCR5$^-$ mCD4 T cells was significantly higher in patients with the presence of specific autoantibodies (Fig. 1D). In comparison, the frequency of MP1 (H1N1)-, MP65 (C.ALB)- and SPIKE (SARS-CoV-2)-specific CD4 T cells was detectable in the blood of all patients (Fig. 1E). Foreign antigen-specific CD4 T cells were also mCD4 T cells. MP1- and SPIKE-specific CD4 T cells expressed PD-1. Neither MP1-, MP65- or SPIKE-specific CD4 T cells expressed significant levels of CXCR5 (Fig. 1E). In conclusion, during AILD, we observed an association between the specific autoimmune humoral response (auto-antibodies; e.g. anti-SLA) and the presence of circulating self-antigen-specific CD4 T cells (e.g. Sepsecs-specific memory CD4 T cells).

### Self-antigen-specific CD4 T cells have a unique transcriptional immune signature during AILD

We sorted self-antigen-specific and foreign antigen-specific mCD4 T cells by flow cytometry for single-cell RNA-seq and TCR-seq analyses (Fig. 2). Unsupervised clustering analysis led to the identification of eight clusters (Fig. 2A). The majority of transcriptional clusters corresponded to distinct antigenic specificities rather than to distinct patients (Fig. 2B and C). The only notable exception was cluster 7, which corresponded to a PDCE2-specific CD4 T cell clonotype from one patient (01–176) with a unique cytotoxic profile (Supplementary Fig. 2). As the memory response against an antigen is generally associated with clonal expansion, we analyzed the diversity and proportion of TCR clonotypes for each antigen reactivity (Fig. 2D). Clonal expansion was observed for every antigenic reactivity, whether associated with self- or foreign-reactivity. Analysis of these expanded clones for cluster affiliation demonstrated that transcriptional clusters were linked to antigenic reactivity (Fig. 2E). Interestingly, all three liver auto-

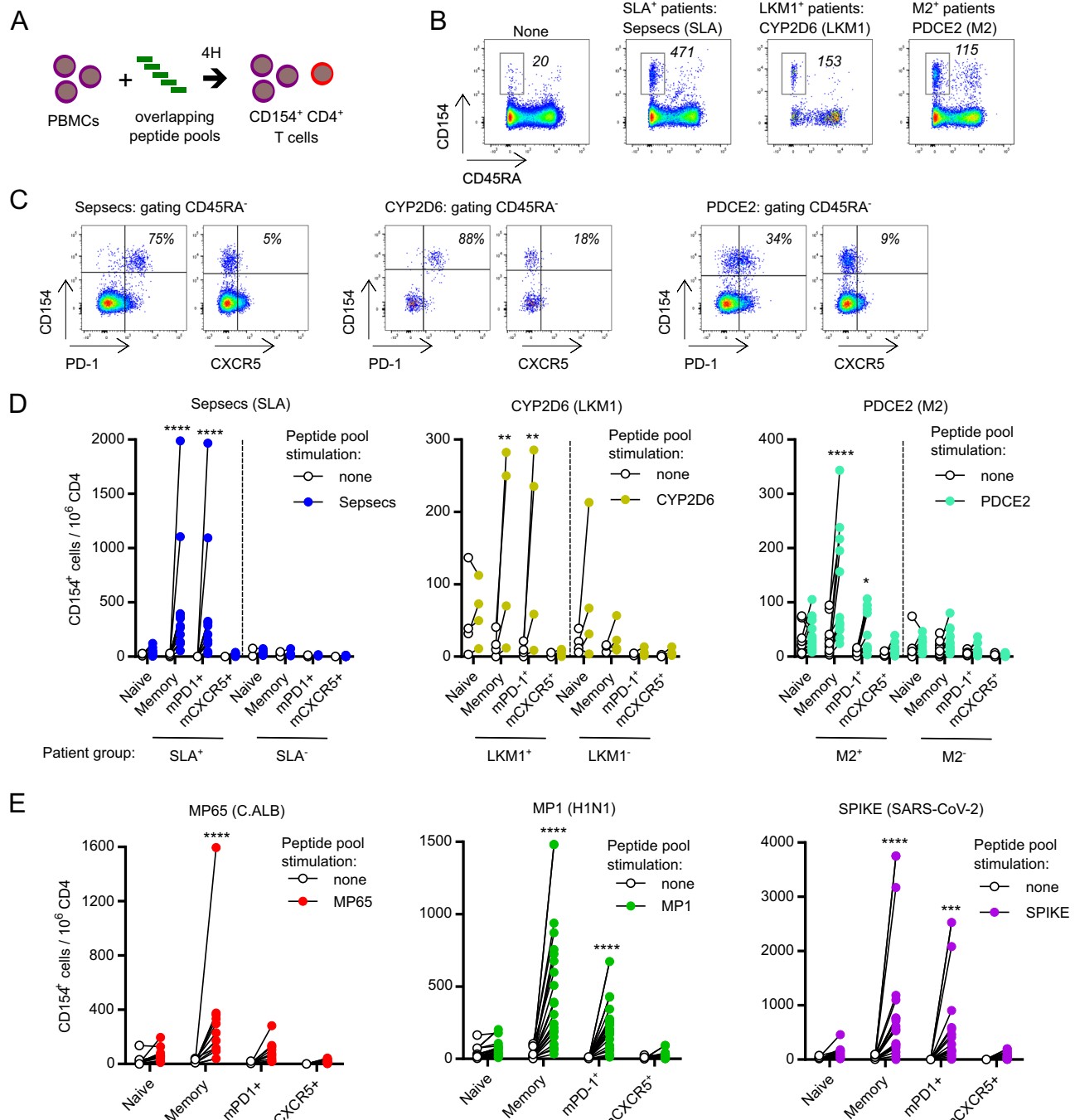

**Fig. 1 | Detection and immune-phenotyping of circulating liver-self-antigen-specific CD4 T cells. A** Schematic representation of the detection of antigen-specific CD4 T cells based on the upregulation of CD40 ligand (CD154) after 4 h of ex vivo peptide stimulation. **B** Pseudocolor dot plot representation of CD45RA and CD154 expression at the surface of CD4 T cells after stimulation with peptide pools from the indicated antigens (numbers indicate mean of CD45RA⁻CD154⁺ cells per million total CD4). **C** Pseudocolor dot plot representation of PD-1, CXCR5, and CD154 expression at the surface of CD45RA⁻ mCD4 T cells after stimulation with peptide pools from the indicated antigens (numbers indicate mean percentage). **D** Frequency of CD154⁺ naïve, memory, memory PD-1⁺ (mPD-1⁺) and memory CXCR5⁺ (mCXCR5⁺) CD4 T cells per million CD4 T cells after Sepsecs, CYP2D6 or PDCE2 peptides stimulation of PBMCs from 12 SLA⁺ patients and 9 SLA⁻ patients (left; ****: $p < 0.0001$); or 4 LKM1⁺ patients and 4 LKM1⁻ patients (central; **: $p = 0.0098$ (Memory); $p = 0.0090$ (mPD1⁺)); or 14 M2⁺ patients and 14 M2⁻ patients (right; ****: $p < 0.0001$, *: $p = 0,0471$). **E** Frequency of CD154⁺ naïve, memory, mPD-1⁺ and mCXCR5⁺ CD4 T cell per million CD4 T cells after MP65 (C.ALB; ****: $p < 0.0001$), MP1 (H1N1; ****: $p < 0.0001$) or SPIKE (SARS-CoV-2; ****: $p < 0.0001$, ***: $p = 0.0003$) peptides stimulation of PBMCs from 13, 24 and 23 patients respectively. Two-sided, two-way ANOVA with Sidak's multiple comparisons test was used for (**D**, **E**). Source data are provided as a Source Data file.

reactivities were grouped in the same clusters. Sepsecs-, CYP2D6- and PDCE2-specific CD4 T cells were in the clusters 2 and 3. MP1-specific CD4 T cells were in the cluster 0; MP65-specific CD4 T cells were in the cluster 5; and SPIKE-specific CD4 T cells were in the cluster 6. Two clusters (4 and 1) were not linked to antigenic reactivity. We annotated

four antigen-reactivity groups based on unsupervised transcriptional clustering: the H1N1 cluster (ex-cluster 0); the C.ALB cluster, (ex-cluster 5); the SARS-CoV-2 cluster (ex-cluster 6); and the auto-reactivity cluster (ex-clusters 2 and 3) (Fig. 2F and G and supplementary Data 2). The gene signature of autoantigen-reactive cells was linked to the

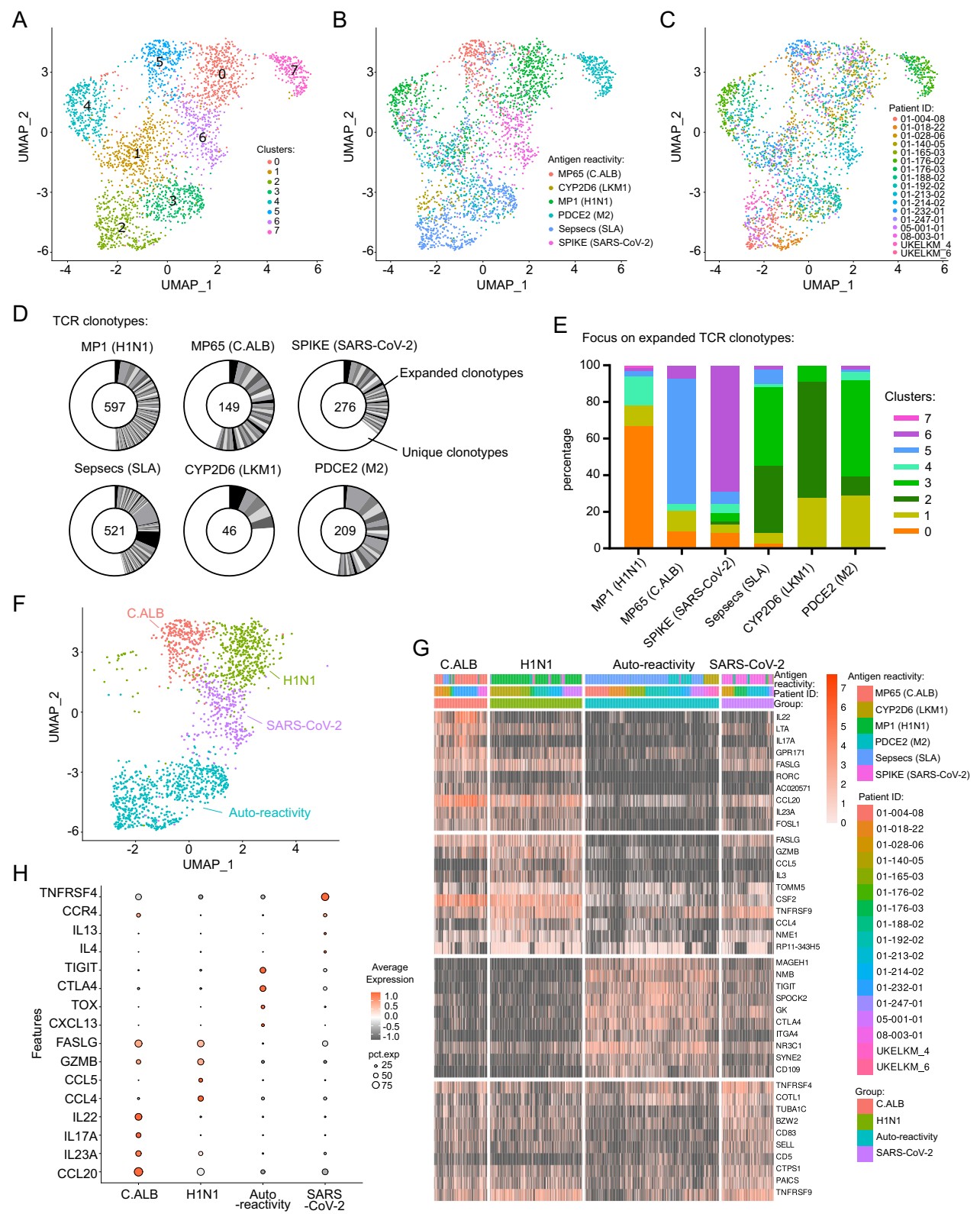

regulation of T cell activation, immuno-exhaustion, and adaptive immune response (B-helper function) (*CTLA-4, TIGIT, BATF, CD74, PRDM1, TOX, IL21, CXCL13*) (Fig. 2H and supplementary Data 2 and 3). In comparison, the H1N1 gene signature was associated with a classical antiviral response and T cell chemotaxis (*GZMB, CCL4, CCL5, and FASLG*). The Candida Albicans gene signature was associated with a type 17 helper response (*CCL20, IL22, IL17A, IL23A*). SARS-CoV-2 gene

signature was associated with a type 2 helper response and B cell activation (*TNFRSF4, CCR4, IL4, and IL13*). We had the opportunity to longitudinally analyze the TCR clonotypes of Sepsecs-specific and MP65-specific CD4 T cells of patients (*n* = 3 and 1, respectively) over 3 years (2018 to 2021). Although variable numbers of novel clones were detected in patients' blood over time, some conserved Sepsecs-specific CD4 T clonotypes were detected over the 3-year period

**Fig. 2 | Single-cell RNA sequencing of liver-self-antigens-specific CD4 T cells.** A total of 2768 CD154$^+$ mCD4 T cells (648 Sepsecs-, 113 CYP2D6-, 583 PDCE2-, 273 MP65-, 798 MP1- and 353 SPIKE-specific CD4 T cells) from 7 SLA$^+$, 3 LKM1$^+$ and 6 M2$^+$ patients were analyzed for transcriptome and TCR sequences at the single cell level using FB5P-seq. **A** UMAP representation of antigen-specific memory CD4 T cell single-cell transcriptomes, colored by non-supervised Louvain cluster identity. **B** UMAP representation colored by antigen reactivity. **C** UMAP representation colored by patient ID. **D** TCRαβ clonal diversity of antigen-specific single T cells for each antigen-reactivity. Numbers indicate the number of single cells analyzed with a TCRαβ sequence. Black and grey sectors indicate the proportion of TCRαβ clones (clonotype common to ≥ 2 cells) within single-cells analyzed; white sector: unique clonotypes. **E** Louvain cluster distribution of antigen-specific T cell clonotypes for each reactivity, as indicated. **F** UMAP representation of selected antigen reactivity cluster groups of memory CD4 T cells (C.ALB, H1N1, SARS-CoV-2, and auto-reactivity). **G** Single-cell gene expression heatmap for top 10 marker genes of cells from each antigen reactivity group defined in (**F**). **H** Dot plot representation of four selected marker genes of clusters C.ALB, H1N1, SARS-CoV-2, and auto-reactivity. Source data are provided as a Source Data file.

(Supplementary Fig. 3), suggesting chronically active clonal expansion of autoreactive CD4 T cells despite their immuno-exhausted molecular profile.

Because we hypothesized a close link between circulating auto-reactive CD4 T cells and reactive CD4 T cells in the tissue, we established gene module scores based on published datasets of classical $T_{RM}$ CD4 T cells in healthy tissue (lung, skin and intestine)[46], CXCL13$^+$ CD4 T cells ($T_{RM-like}$ CD4 T cells) in the liver during Hepatitis B Virus infection (HBV)[41] or during Hepatocellular carcinoma (HCC)[40] and $T_{PH}$ cell in synovial fluids during autoimmunity[30,56] (Supplementary Data 4). Unfortunately, no complete transcriptomic data of classical liver $T_{RM}$ CD4 T cells from healthy liver were available in the literature. In contrast to classical $T_{RM}$ CD8 T cells, CD103 is not a specific marker for $T_{RM}$ CD4 T cells[9,45–47,57–59]. We observed the circulating liver-autoreactive CD4 T cells had significantly high $T_{PH}$ and $T_{RM-like}$ scores (CXCL13$^+$ CD4 T cells in the liver during HBV infection and HCC). Limited overlap was observed with the classical $T_{RM}$ CD4 T cell signature from healthy tissue (Supplementary Fig. 4 and Supplementary Data 5).

Altogether, these data demonstrate that distinct circulating liver-self-antigen-specific CD4 T cells had a common B-helper (adaptive immune response) signature, with high-level expression of immune checkpoint (immuno-exhausted transcriptional profile) and strong overlap with transcriptional signatures of $T_{PH}$ cells and $T_{RM-like}$ CD4 T cells under chronic activation in tissues.

The B-helper signature of liver-autoreactive CD4 T cells is consistent with the fact that tertiary lymphoid structures may be found in the liver during auto-immunity[60]. Accordingly, we performed spatial multi-phenotyping analysis of a liver biopsy from one AIH patient and identified tertiary lymphoid structures (TLS) characterized by the presence of CD20$^+$ B-cells cells, CD21$^+$ follicular dendritic cells and PD-1$^+$ CD4 T cells (Supplementary Figs. 5 and 6). The TLS was enriched in CXCR5$^+$ cells suggesting germinal center formation and/or B cell retention.

## Clonal overlap between circulating autoreactive and intra-hepatic CD4 T cells in AILD

We then attempted to answer the following question: how related to liver infiltrating CD4 T cells are circulating autoreactive CD4 T cells in AILD? We hypothesized that circulating autoreactive CD4 T cells might represent clonal relatives of liver CD4 T cells and that their specific molecular profile was induced during chronic antigen reactivity in the tissue. For one SLA$^+$ patient analyzed by peptide restimulation and single-cell sequencing (Fig. 2, patient 01–192), we also had access to a frozen liver biopsy, from which we derived bulk TCR-beta-seq data (Supplementary Fig. 7 and supplementary Data 6). Six out of sixty (10%) Sepsecs-specific circulating CD4 TCR clonotypes were also detected in the liver, compared to only one out of forty-six (2.2%) MP1-specific TCR clonotypes (H1N1). Bulk TCR-beta sequence counts corresponding to Sepsecs-specific clonotypes suggested few clonal expansions dominating the Sepsecs-specific response also in the liver (Supplementary Fig. 7). Thus, for this patient, circulating self-antigen-specific CD4 T cells reflected clonally expanding autoantigen-specific T cells within the liver.

For most AILD patients, no self-antigen reactive TCRs were known. Therefore, to continue exploring the relationship between

circulating and liver CD4 T cells, we took advantage of our initial observation that in AILD, most circulating self-antigen-specific CD4 T cells have a PD-1$^+$CXCR5$^-$CD45RA$^-$ memory phenotype[22] (Fig. 1). We had access to paired blood-liver samples from 4 AILD patients; for each, we extracted genomic DNA and performed bulk TCR-beta-seq from blood PD-1$^+$CXCR5$^-$ mCD4 T cells, blood PD-1$^-$ mCD4 T cells, and whole liver biopsy (Supplementary Fig. 8, Supplementary Fig. 9A and supplementary Data 6 and 7). First, we observed that the 100 most abundant TCR-beta sequences in the liver accounted for more than 50% of intrahepatic TCRs, suggesting strong local clonal expansion (Supplementary Fig. 9B). We also observed that the PD-1$^+$CXCR5$^-$ mCD4 T cell population shared more TCR-beta sequences with the liver than PD-1$^-$ mCD4 T cells (Supplementary Fig. 9C). Interestingly, the proportion of cells within the top 100 liver TCR-beta sequences was higher in the PD-1$^+$CXCR5$^-$ mCD4 T cell population, than in the PD-1$^-$ mCD4 T cells (Supplementary Fig. 9D). This data demonstrated the circulating PD-1$^+$CXCR5$^-$ mCD4 T cell population contains dominant intrahepatic CD4 T cell clonotypes.

Based on this, we performed droplet-based single-cell RNA-seq combined with TCR sequencing of blood PD-1$^+$CXCR5$^-$ mCD4 T cells from the four patients to characterize the transcriptional profile of cells with TCR clonotypes detected in the matched liver biopsies (Fig. 3A). First, cluster analysis of PD-1$^+$CXCR5$^-$ mCD4 T cells revealed the heterogeneity of these cells (Fig. 3B and supplementary Data 8). The main subsets identified by unsupervised azimuth annotation and marker gene analysis were: *GZMK*$^+$ effector CD4 T cells ($T_{EM}$ GZMK$^+$, cluster 1); *HLA-DR*$^+$ activated central memory CD4 T cells ($T_{CM}$ HLA-DR$^+$, cluster 2), *NKG7*$^+$ effector CD4 T cells ($T_{EM}$ NKG7$^+$, cluster 3); cytotoxic (*GZMB*$^+$*GNLY*$^+$) CD4 T cells (CTL, cluster 5) and *FOXP3*$^+$ (regulatory) CD4 T cells ($T_{REG}$, cluster 6) (Fig. 3B and C, and Supplementary Figs. 10 and 11). Clusters 4, 7, and 8 were likely subsets of $T_{CM}$ but were difficult to further annotate based on marker genes.

In this data set, we tracked the top 100 liver TCR-beta sequences (Fig. 3D). For each of these liver TCR-beta clones, we assigned a cluster identification based on single-cell transcriptomic data from PD-1$^+$CXCR5$^-$ mCD4 T cells (Fig. 3D). The analysis revealed that the top expanded intrahepatic clones found in the blood were mainly distributed between cluster 1 ($T_{EM}$ GZMK$^+$), cluster 2 ($T_{CM}$ HLA-DR$^+$) and cluster 3 ($T_{EM}$ NKG7$^+$). We computed gene set scores based on our analyses of peptide-stimulated T cells (Fig. 2 and supplementary Data 2), with the expression of the top 50 genes (avg_log2FC) from auto-reactivity or H1N1 reactivity, respectively. These scores were then quantified in the droplet-based scRNA-seq dataset of non-stimulated PD-1$^+$CXCR5$^-$ mCD4 T cells (Fig. 3E). Cells in cluster 2 ($T_{CM}$ HLA-DR$^+$) had the highest score for the auto-reactivity module, while cells in clusters 3 and 5 had the highest score for the H1N1 module (Supplementary Data 9). These data suggest that cells in cluster 2, PD-1$^+$CXCR5$^-$HLA-DR$^+$ $T_{CM}$, were the main circulating CD4 T cell clonal counterpart of liver-autoreactive CD4 T cells. Interestingly, this cluster 2 also contained proliferative CD4 T cells, reinforcing the idea of local clonal expansion (Supplementary Fig. 11).

As performed before, we used gene module scores of classical $T_{RM}$ CD4 T cells from healthy lung, skin, and jejunum, liver $T_{RM-like}$ CD4 T cells during HBV infection or HCC, and of $T_{PH}$ cells from synovial fluid during autoimmunity. The module score analysis revealed that cells in

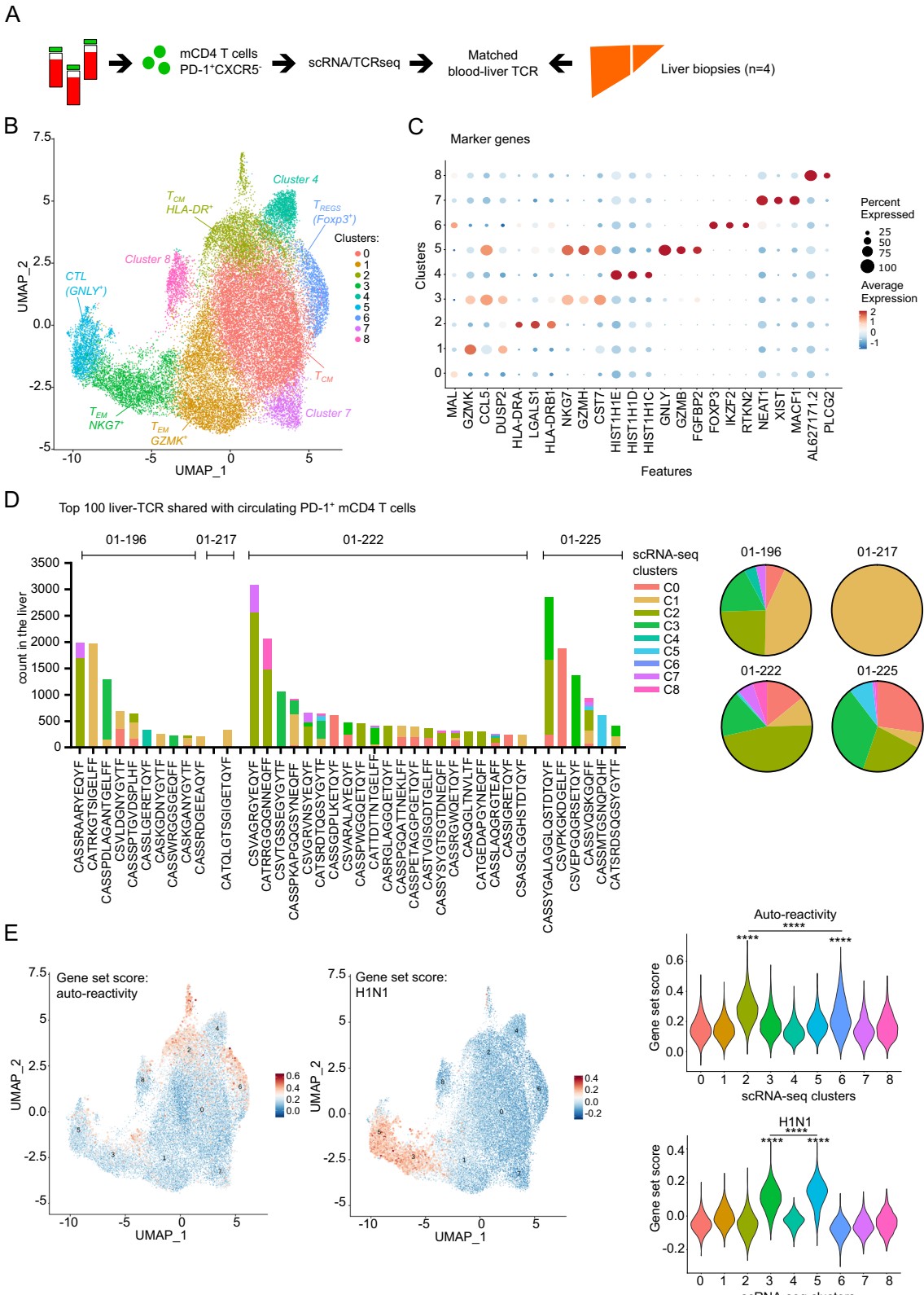

cluster 2 had the highest score for the $T_{PH}$ module (Supplementary Fig. 12 and Supplementary Data 4). Cluster 2 had also significant enrichment scores for the lung $T_{RM}$ CD4 T cell module and for the $T_{RM-like}$ CD4 T cell module although these signatures were not exclusive of the cluster 2 (Supplementary Data 9). The data reinforces the idea that cluster 2 contains circulating liver-autoreactive CD4 T cells which may be imprinted by the chronic antigen-activation in the tissues.

We further sought to relate the transcriptional signatures of peptide-stimulated and untouched circulating autoreactive mCD4 T cells (Figs. 2 and 3). Based on previously published results[22], we synthesized HLA-DRB1 03:01 biotinylated monomers loaded with the Sepsecs187-197 epitope peptide, assembled fluorescent-labeled pMHCII tetramers, and monitored Sepsecs-specific CD4 T cells in the blood of six SLA+ HLA-DR3+ patients (Fig. 4A). Peptide-MHCII tetramer positive

**Fig. 3 | Tracking TCR clonotypes between liver biopsies and circulating CD4 T cell subsets. A** Experimental design for scRNA-seq and TCR-seq of memory PD-1⁺CXCR5⁻ CD4 T cells from four distinct AILD patients. **B** UMAP representation of circulating memory PD-1⁺CXCR5⁻ CD4 T cell transcriptomes, colored by non-supervised Louvain clustering. **C** Dot plot representation of top 3 marker genes of each clusters in (**B**). **D** Frequency representation ('counts') of each top 100 largest liver TCRβ sequences found in circulating PD-1⁺ mCD4 T cells. For each TCRβ sequences the cluster affiliation is indicated based on scRNA-seq and TCR-seq of

memory PD-1⁺CXCR5⁻ CD4 T cells. Pie chart represent the global cluster affiliation of the top 100 largest liver TCRβ sequences found in circulating PD-1⁺ mCD4 T cells per patient. **E** Left: UMAP representation of circulating memory PD-1⁺CXCR5⁻ CD4 T cell transcriptomes colored by gene set score for the indicated antigen-reactivity module. Right: violin plots of gene set score distribution of cells from different Louvain clusters, as indicated. Two-sided, pairwise comparison with a paired Wilcoxon rank test. All *p* values are listed in the supplementary Data 9 for (**E**). Source data are provided as a Source Data file.

(TTpos) CD4 T cells were detectable and displayed a PD-1⁺ memory phenotype (Fig. 4B and Supplementary Fig. 13). We performed plate-based single-cell transcriptomic analyses of pMHCII TTpos Sepsecs-specific CD4 T cells. We compared pMHCII TTpos and TTneg CD4 T cells for their expression of gene modules associated with antigen reactivities defined in Fig. 2 (Fig. 4C) and in the PD-1⁺ CXCR5⁻ mCD4 T cell subsets identified in Fig. 3 (Fig. 4D), and also in CD4 T cells from distinct healthy or pathogenic tissues, as mentioned before (Supplementary Fig. 14). As expected, pMHCII TTpos CD4 T cells expressed significantly higher levels of gene modules associated to self-antigen-specific CD4 T cells (Fig. 4C) and cluster 2 T_CM HLA-DR⁺ CD4⁺ cells (Fig. 4D). We also observed that pMHCII TTpos CD4 T cells expressed significantly higher levels of gene modules associated to the tissue immune response (T_PH and T_RM-like CD4 T cells); confirming the link between circulating autoreactive T cells and liver autoimmunity.

Altogether, our integrative analyses of circulating self-antigen-specific and dominant intrahepatic CD4 T cell clones in AILD patients converge to establish that some autoreactive CD4 T cell clones can be found both in the liver and in the blood, where they are defined as PD-1⁺CXCR5⁻HLA-DR⁺ mCD4 T cells, with a specific autoimmune and immuno-exhausted transcriptional profile related to an active immune response in the tissue.

## High-dimensional phenotyping of circulating liver-autoreactive CD4 T cells

We then examined the precise phenotype of these circulating liver-autoreactive CD4 T cells by spectral flow cytometry using PBMCs from patients affected by different liver diseases: five control non-autoimmune patients with non-alcoholic steatohepatitis (NASH), fourteen AIH patients with an active disease, and thirteen AIH patients in remission under treatment (Supplementary Table 1). We focused on the mCD4 T cell population and performed clustering analysis (Fig. 5A–D). The comparison revealed the significant increase of three cell clusters in patients with active AIH in comparison to NASH patients (C6, C12, and C15, Fig. 5B). The C6 and C15 clusters were T_CM CD27⁺TIGIT⁺PD-1⁺CXCR5⁻CD49d⁺CD25⁻CD127⁻ICOS⁺ CD4 cells and expressed CD38, a marker we have already observed upregulated by CD4 T cells during active AIH[22] (Fig. 5C and D, Supplementary Fig. 15). The C15 cluster was characterized by high HLA-DR expression and may represent the liver-autoreactive CD4 T cells identified previously (Fig. 3). This cluster (C15) was high in the blood of active AIH patients compared to AIH patients in remission under treatment (Supplementary Fig. 15). The C12 cluster was characterized by CD57 expression but its frequency was very low. We performed a supervised analysis of the HLA-DR⁺ subset and confirmed that it was significantly more frequent during the active phase (AIHa) than in patients with NASH or in remission (AIHr) (Supplementary Fig. 16). In parallel, in four active AIH patients, we performed the intracellular analysis of the PD-1⁺TIGIT⁺CXCR5⁻ CD4 subset with a new panel of antibodies. We confirmed its heterogeneity; this subset consisted of classical T_REG (FOXP3⁺CD127⁻EOMES⁻), cytotoxic cells (GZMB⁺HLA-DR⁻EOMES⁻GZMA⁺) and HLA-DR⁺ FOXP3⁻EOMES⁻ cells (Supplementary Fig. 17). This was in agreement with the transcriptomic data (Fig. 3) to demonstrate that the liver-autoreactive CD4 T cell cluster (HLA-DR⁺PD-1⁺) was different from the T_REG and cytotoxic (CTL) PD-1⁺ CD4 T cell clusters.

Thus, after exclusion of T_REGs (using CD127 and CD25 or FOXP3 expression), HLA-DR expression emerged as the most relevant marker for delineating liver-autoreactive CD4 T cells among PD-1⁺TIGIT⁺CD4 T cells.

To get an insight into the functionality of these rare cells, we used this strategy of T_REG exclusion to sort HLA-DR⁺PD-1⁺TIGIT⁺CXCR5⁻ mCD4 T cells and to perform their TCR stimulation in vitro for bulk-RNA sequencing (Fig. 5E and F, Supplementary Fig. 18 and supplementary Data 10). Like autoreactive CD4 T cells after peptide stimulation, HLA-DR⁺PD-1⁺TIGIT⁺CXCR5⁻ mCD4 T cells significantly upregulated *CXCL13, IL21, CTLA-4, PRDM1, TIGIT, MAF,* and *TOX,* upon TCR stimulation. These cells also expressed *LGALS1, CXCR3, HLA-DR, and CD74* like CD4 T_CM HLA-DR⁺ cells; these genes were downregulated by TCR stimulation. They expressed genes associated to the T_PH and T_RM-like CD4 cell signatures like *ENTPD1 (CD39), TIM3, CXCR6, CCR5,* and *CCR2* that linked them to the tissue immune response. They could also express *TNF* and *INFG,* but not *IL2,* after TCR stimulation. This subset presented high proximity with autoreactive CD4 T cells and circulating dominant intrahepatic clonotype, thus, we can conclude that HLA-DR⁺PD-1⁺TIGIT⁺CXCR5⁻ CD4 T cells contained the liver-autoreactive CD4 T cells.

## The transcriptional signature of liver-autoreactive CD4 T cells is dictated by the hepatic environment

Since circulating liver-autoreactive CD4 T cells have a specific transcriptional signature in humans, we investigated whether their molecular profile was due to antigen reactivity in the hepatic environment or to a lineage-specific signature. To answer this question, we designed a non-TCR-transgenic mouse model that allows the analysis of the emergence and modulation of CD4 T cell responses against a model antigen expressed by hepatocytes. We used a model of Balb/c mice in which the hemagglutinin (HA) expression was restricted to hepatocytes, by an inducible Cre recombinase under the control of hepatocyte-specific mTTR (transthyretin) promotor (HA/iCre) (Fig. 6A and Supplementary Fig. 19). Tamoxifen induced HA expression in the liver of HA/iCre mice but did not induce an anti-HA humoral response (Fig. 6A–C). Intramuscular injection of a Cre-coding adenovirus (AdCre) or an HA-encoding plasmid was performed to generate peripheral immunization against the HA antigen (HA i.m, Fig. 6A and C). HA-specific memory (CD44^high) CD4 T cells were detected with pMHCII tetramers in the spleen but not in the liver of mice after peripheral immunization (Supplementary Fig. 20).

To analyze HA-specific CD4 T cells in the liver, we induced HA expression in the liver, by using tamoxifen in pre-immunized HA/iCre mice (HA i.m + Tamoxifen, Fig. 6A); we used littermates lacking liver-restricted expression of an inducible Cre recombinase (Ctrl) as controls (Supplementary Fig. 19). In this condition, we detected HA-specific CD4 T cells in the liver of immunized mice expressing HA by hepatocytes after tamoxifen treatment (Fig. 6D), showing active recruitment from the periphery of HA-specific memory CD4 T cells into the liver following local antigen expression. Liver HA-specific CD4 T cells expressed high levels of PD-1 (Fig. 6E).

In this mouse model, we analyzed the transcriptional profile of HA-specific CD4 T cells in mouse liver and spleen. Overall, all spleen HA-specific CD4 T cells, irrespective of liver HA expression, clustered

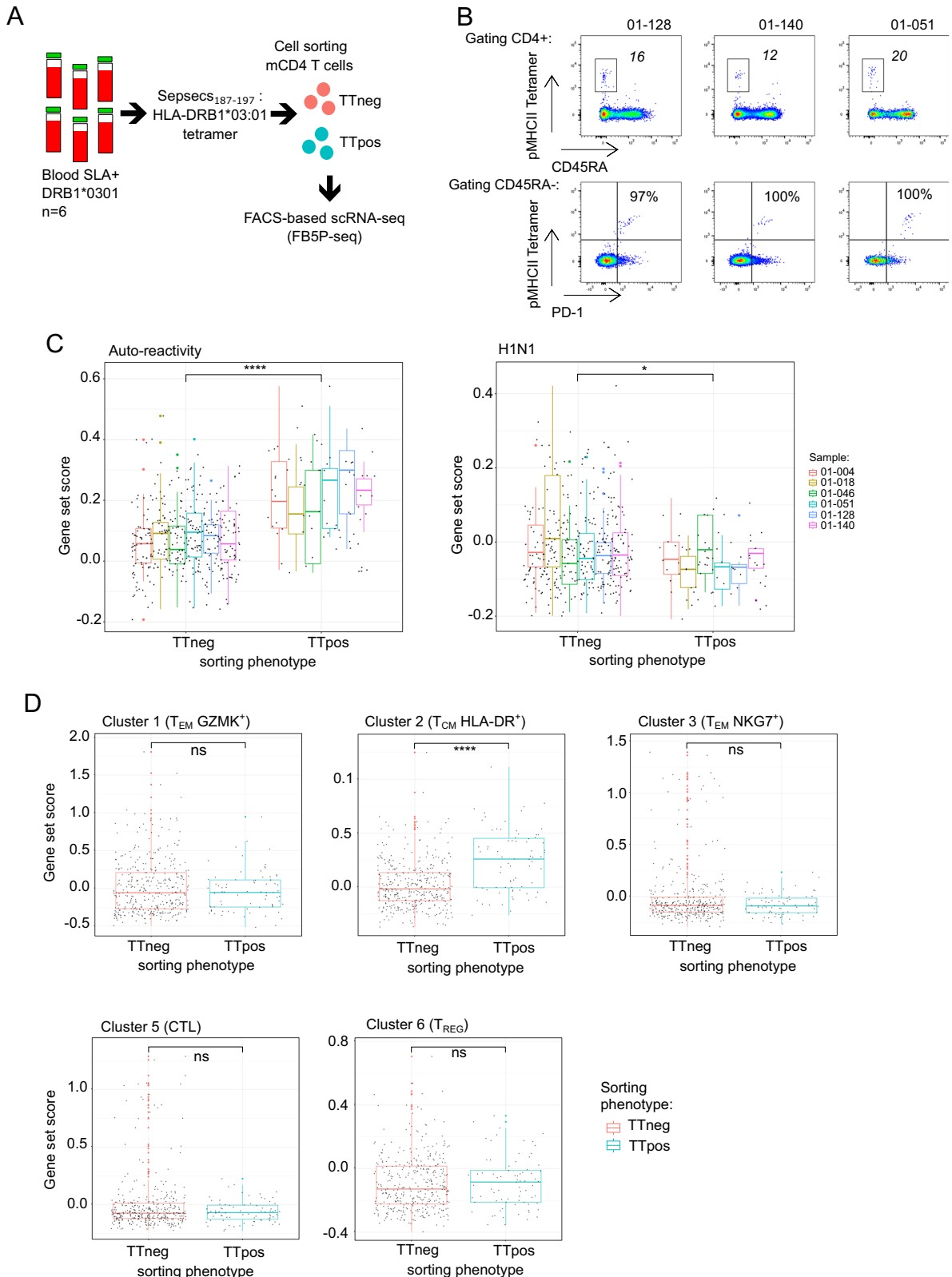

together, whereas liver HA-specific CD4 T cells show a distinct transcriptional signature (Fig. 6F and G and supplementary Data 11). Besides expressing genes associated with the inflammatory response (e.g. *Ox40*, *Cd137*, *Ccl5,* and *Icos*), liver HA-specific CD4 T cells expressed genes involved in the regulation of proliferation and activation (e.g. *Tigit*, *Ctla4*, *Prdm1,* and *Casp3*) (Fig. 6H). Comparative analysis with human liver-self-antigen specific transcriptional signature revealed four common pathways linked to the regulation of T cell activation (GO:0042110 T cell activation; GO:0002694 reg. of leukocyte activation; GO:0051249 reg. of lymphocyte activation; GO:0050863 reg. of T cell activation; supplementary Data 3). The analysis of the genes from these pathways revealed common genes shared between liver HA-specific CD4 T cells in mice and circulating liver-autoreactive CD4 T cells in humans (Fig. 6I). Both cell types

**Fig. 4 | Single cell transcriptomic analysis of unstimulated Sepsecs-specific CD4 T cells. A** Experimental design for scRNA-seq data analysis of Sepsecs/SLA$_{185-197}$ HLA-DRB1*0301 pMHCII tetramer positive cells. **B** Pseudocolor dot plot representation of pMHCII tetramer staining in three of six patients. Top: surface expression of CD45RA and pMHCII tetramer in CD4$^+$ T cells. Bottom: surface expression of PD-1 and pMHCII tetramer in CD4$^+$ CD45RA$^-$ T cells (numbers indicate number of CD45RA$^-$Tetramer$^+$ cells per million total CD4 and percentage). **C** Box plots of gene set score distribution of pMHCII tetramer negative (TTneg, $n = 366$) or

positive (TTpos, $n = 75$) cells from six AIH patients for antigen reactivity modules as indicated (auto-reactivity, $p = 6.4 \times 10^{-14}$; H1N1, $p = 0.011$). Each box plot indicates a patient sample. **D** Box plots of gene set score distribution of pMHCII tetramer negative (TTneg) or positive (TTpos) cells for top marker genes of circulating memory PD-1$^+$CXCR5$^-$ CD4 T cell clusters defined in Fig. 3E (Cluster 1, $p = 0.51$; Cluster 2, $p = 1 \times 10^{-9}$; Cluster 3, $p = 0.37$; Cluster 5, $p = 0.54$; Cluster 6, $p = 0.64$). Data are presented as mean values ± SD. Two-sided, Unpaired Mann-Whitney test was used. Source data are provided as a Source Data file.

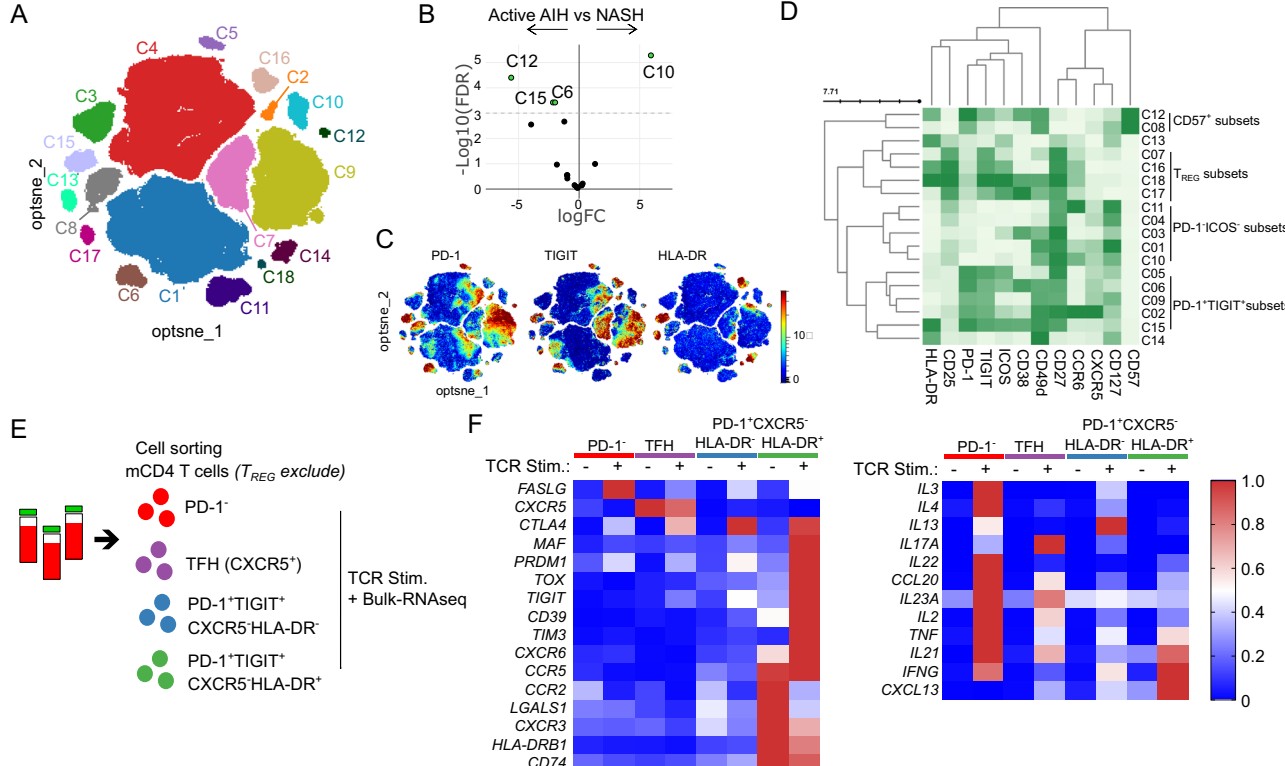

**Fig. 5 | Characterization of circulating PD-1$^+$TGIT$^+$HLA-DR$^+$ non-T$_{REG}$ CD4 T cells as liver-autoreactive CD4 T cells. A** opt-SNE representation of blood memory CD4 T cell subsets, colored by non-supervised FlowSOM clustering, from 5 control non-autoimmune patients with non-alcoholic steatohepatitis (NASH), 14 AIH patients with an active disease (Active AIH), and 13 AIH patients in remission under

treatment. **B** Differential cluster abundance between active AIH and NASH patients. **C** PD-1, TIGIT, and HLA-DR expression. **D** Surface marker heatmap of the clusters identified in (**A**). **E** Experimental design for bulk-RNA-seq after cell sorting and 24 h in vitro TCR stimulation. **F** Selected genes heatmap of indicated CD4 T cell subsets after or not TCR stimulation from five distinct patients (mean representation).

expressed high levels of *TIGIT, CTLA-4, BATF,* and *PRDM1*. These data demonstrated that immuno-regulatory genes were induced during local immune responses in the liver. This suggests that the immune signature of human circulating liver-autoreactive CD4 T cells may be imprinted by the hepatic environment, with these cells leaking out of damaged tissue after a phase of high reactivity in the tissue.

### Immune checkpoint molecules control antigen-specific hepatitis in a CD4 T cell-dependent manner

The upregulation of immune checkpoint molecules by antigen-specific CD4 T cells after their reactivity in the liver suggested the potential regulation of these cells by the hepatic environment. Therefore, we blocked PD-1 and CTLA-4 pathways, with specific immune checkpoint inhibitors (ICI) in our mouse model of hepatic infiltration by HA-specific CD4 T cells, after immunization and induction of HA expression by hepatocytes (Fig. 7A). The data revealed a significant hepatitis marked by high histological liver inflammation score when mice receiving ICI were pre-immunized and expressed HA in the liver (HA/iCre + ICI, Fig. 7B and C). Moreover, PD-1 and CTLA-4 blockade was associated with higher number of IFNγ-secreting CD4 and CD8 T cells in the spleen and in the liver after HA peptide stimulation in vitro

(Supplementary Fig. 21). Mice immunized but unable to express HA in their liver after tamoxifen treatment did not show significant liver inflammation after PD-1 and CTLA-4 blockade (Ctrl + ICI, Fig. 7B and C). These data demonstrate that the hepatic environment specifically regulates the antigen-specific response by engaging immune checkpoint receptors.

Then, we assessed the role of CD4 T cells in this response by depleting CD4$^+$ cells with anti-CD4 antibodies in immunized HA/iCre mice treated with tamoxifen and PD-1/CTLA-4 blocking antibodies (Fig. 7D). First, CD4 depletion was associated with significantly lower immunization rate according to anti-HA antibody levels in the serum (Fig. 7E). Interestingly, histological analysis revealed significantly reduced liver damages in CD4 depleted mice, showing the major role of CD4 T cells in promoting antigen-specific liver inflammation in our model (Fig. 7F and G). The hepatic inflammation observed after PD-1 and CTLA-4 blockade was linked to substantial HA-specific CD8 response in the spleen and in the liver, but depletion of CD4 T cells before the induction of HA expression in the liver abrogated the HA-specific CD8 response (Fig. 7H). These results demonstrate that the hepatic environment quickly regulates local immune response against an antigen expressed by hepatocytes through upregulation of immune

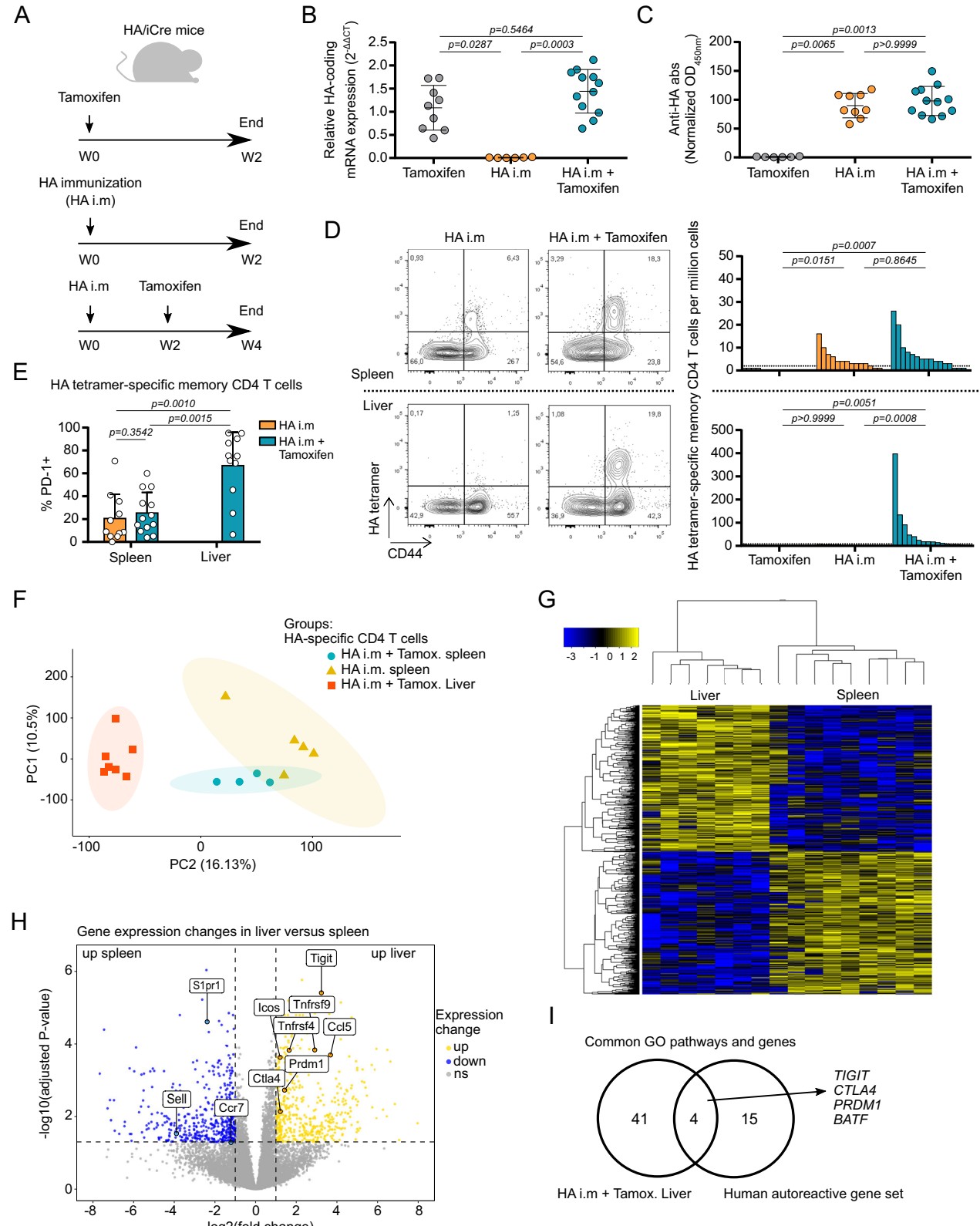

checkpoint pathways in a CD4-dependent manner. This data revealed a major role of the liver-antigen-specific CD4 T cells in the initiation of immune response against a hepatic antigen.

## Discussion

In this study, we analyzed the transcriptome, TCR repertoire, and single-cell protein expression of circulating liver-self-antigen-specific CD4 T cells and expanded intrahepatic CD4 T cell clonotypes found in the blood of AILD patients. We have identified a peripheral subset of liver-autoreactive CD4 T cells (non-T$_{REG}$ T$_{CM}$ CXCR5$^-$CD49d$^+$PD-1$^+$TIGIT$^+$ICOS$^+$HLA-DR$^+$CD38$^+$ CD4 T cells) with an autoimmune transcriptional profile (*CXCL13, IL21, TIGIT, CTLA4, PRDM1, TOX, HLA-DR, CCR2, ENTPD1, LGALS1, CXCR3, CXCR6, TIM3*) that is linked to clonal expansion in the liver. Using a mouse model to track the early immune

**Fig. 6 | Analysis of antigen-specific CD4 T cells from the liver and the spleen of an in vivo non-TCR-transgenic mouse model. A** Experimental design for investigation of immune response against the HA antigen in the liver. HA/iCre mice received either tamoxifen treatment ($n = 9$), HA immunization treatment (HA i.m; $n = 15$), or HA immunization followed 2 weeks later by tamoxifen treatment (HA i.m + Tamoxifen; $n = 12$). Mice were euthanized 2 weeks after the last treatment. **B** Relative HA mRNA expression in the liver. ACTB was used as loading control. **C** Analysis of normalized anti-HA antibody rate in serum of mice. **D** Analysis of HA-tetramer-specific CD4 T cells in the spleen (top) and in the liver (bottom). Contour plot representation of HA tetramer staining and CD44 expression in CD4 T cells from HA i.m and HA i.m + Tamoxifen mice (left). Frequency of HA-tetramer-specific memory (CD44$^{high}$) CD4 T cells per million cells (right). Dotted lines in the

frequency graphs represent threshold of positive detection of HA-specific CD4 T cell population. **E** Analysis of PD-1 expression in detectable HA-specific memory CD4 T cells from the spleen and the liver of HA i.m and HA i.m + Tamoxifen mice. **F** Principal component analysis (PCA) of bulk-RNA-seq transcriptome of HA-specific CD4 T cells isolated from the liver or the spleen. **G** Gene expression heatmap of HA-specific CD4 T cells. **H** Volcano plot of gene expression change between liver and spleen HA-specific CD4 T cells. **I** Genes from GO enrichment pathways upregulated and shared between mouse liver HA-specific CD4 T cells and human autoreactive CD4 T cells. Data are presented as mean values ± SD in graphs (**B**, **C**, and **E**). Dunn's multiple comparisons test was used for (**B**–**D**). Two-sided Mann-Whitney test was used for (**E**). $p$-values and adjusted $p$-values (multiple comparisons) are indicated. Source data are provided as a Source Data file.

response against an antigen conditionally expressed in the liver, we observed that liver CD4 T cells targeting this antigen acquired an immuno-exhausted (*TIGIT, PRDM1, CTLA-4*) transcriptomic signature comparable to that of circulating liver-autoreactive CD4 T cells in AILD patients, and are essential for inducing a complete immune response against a liver antigen responsible for liver damage. Our data support the concept that circulating CD4 T cells targeting hepatic antigens are liver-derived autoreactive CD4 T cells and that their transcriptomic signature is imprinted by chronic antigen activation in the tissue during autoimmunity. The ability to identify a subset of T cells linked to tissue auto-reactivity in blood opens important prospects for the development of biomarkers and therapies. This is in line with the capacity to identify tumor-specific CD8 T cells in blood which share similarities with their counterparts in tissues, and could therefore be an alternative source for identifying antitumor T cell reactivity[61].

This study also represents a complete deep phenotyping and transcriptome analysis of CD4 T cells targeting hepatic autoantigens in human AILD. We analyzed the transcriptional signature of the CD4 T cell reactivity against three distinct liver-self-antigens and against three common foreign antigens. Upon in vitro peptide stimulation, autoreactive CD4 T cells present a common transcriptomic signature although the three self-antigens have no molecular similarity (Sepsecs, CYP2D6, and PDCE2). Interestingly, this transcriptomic signature is distinct from the anti-fungus Th17 (Candida Albicans reactivity) and the anti-viral (influenza H1N1 and Sars-CoV-2) immune signatures. In a previous publication, we had shown that Sepsecs-specific CD4 T cells, versus CD4 T cells reactive against the Candida Albicans, have a B-helper / pro-inflammatory immune signature[22]. In this new study, we reinforce the data, and we demonstrate that the immune signature of liver autoimmunity is clearly distinct from diverse foreign antigen immune responses. The major immune genes characterizing this signature are *TIGIT, CTLA4, NR3C1, TOX, CXCL13, STAT3*, and *PRDM1*. This autoreactive signature shares a high similarity with the published signature of the $T_{PH}$ CD4 cell subset described in the tissue and blood of other autoimmune diseases[23,24,29–32,38]. However, the lack of data available at the single-cell level for other self-antigen-specific CD4 T cells after short peptide stimulation makes it difficult to identify genes specifically linked to hepatic autoimmunity in our dataset.

Interestingly, circulating liver-autoreactive CD4 T cells have a profile similar to that of tissue-infiltrating ($T_{RM-like}$) CD4 T cells found in various chronic antigen immune contexts, including the liver and other tissues. Expression of PD-1, CXCL13, IL21, TIGIT, PRDM1, TOX, CD39, TIM3, CTLA-4, CXCR3, CCR2, HLA-DR, and CXCR6 is commonly shared with the signature of $T_{PH}$ CD4 cells in the synovium of RA patients, and of $T_{RM-like}$ CD4 T cells like neoantigen-reactive CD4 T cells in metastatic human cancers, tumor-infiltrating CD4 T cells, liver-infiltrating CD4 T cells in HBV-infected and HCC patients, and liver-resident CD4 T cells[23,30,39–41,44,56–58,61–65]. Although these $T_{RM-like}$ signatures described in the literature present overlap with classical $T_{RM}$ CD4 T cells in healthy tissues, it is still difficult to make the distinction between tissue residency and chronic activation, especially during cancer, autoimmunity, and virus immune response[42,43,47,48]. In our

study, some similarities were observed with liver-resident CD4 T cells (PD-1 and CXCR6 expression) although no transcriptomic data were available[57–59]. The main difference is that circulating liver-autoreactive CD4 T cells do not express CD69, which is probably downregulated after tissue egress. Altogether, these observations support the idea that local chronic antigenic reactivity induces a significant modification of CD4 T cells (including residency signature), characterized by the induction of the expression of immune checkpoint molecules and other residency markers (CXCR6). However, comparison of the transcriptomic signature of circulating autoreactive T cells with the transcriptomic signature of distinct $T_{RM-like}$ CD4 T cells revealed a stronger match with the autoreactive T cells in the tissue (the $T_{PH}$ cells) than with CD4 T cells infiltrating the liver during HBV infection or HCC or infiltrating other tissues. This suggests a distinct modification of CD4 T cells depending on their tissue environment and type of antigen reactivity (virus response, cancer, and autoimmunity). Further work will be needed to distinguish the specific transcriptional divergences of CD4 T cells in tissues in relation to their localization, antigen reactivity, and immune environment.

In our study, the markers found in the peripheral blood (e.g. PD-1, TIGIT, and HLA-DR) may be useful for designing biomarkers related to tissue aggression. Their presence in the blood stream can be explained by the fact that during the active phase of the disease, the high clonal expansion in the liver may favor the recirculation of autoreactive T cells. This concept is supported by the fact that the frequency of liver-autoreactive CD4 T cells in the blood of patients decreases during the remission phase, which is generally associated with a reduction in intrahepatic lymphocyte activity[66].

The most common feature shared between liver-autoreactive CD4 T cells and tissue infiltrating CD4 T cells is their high expression of immune-regulatory molecules (e.g. PD-1, TIGIT, CTLA-4), which may be linked to local chronic antigen stimulation and exhaustion. During cancer, the blockade of PD-1 can restore the activity of tumor-infiltrating CD4 T cells demonstrating their local immune involvement and that immune checkpoint molecules control local reactivity[39,63]. In our study, we demonstrate that immune checkpoint molecules in the liver participate in the local control of antigen-specific liver damage which is dependent of the CD4 T cell response. In humans, hepatic lesions can be induced by anti-cancer immunotherapy (anti-PD-1 and anti-CTLA-4) and are associated with a predominant CD8 T cell infiltrate in the lobular zone, suggesting a reactivation of the local liver-antigen-specific T cell response[67–69]. The major difference with AIH is that during immunotherapy-induced acute hepatitis, no signs of chronicity are reported, and the acute hepatitis is rapidly controlled by immunosuppressive treatments without relapse after treatment withdrawal. In contrast, AIH patients' liver biopsies show the complete architecture of a chronic adaptive immune response at diagnosis, which is usually late in the course of the disease. A large number of B cells and of CD4 T cells are observed in the portal zone[68,69]. The detection of tertiary lymphoid structures in the portal zone in our study supports the idea of a prolonged activation period to achieve such immune organization.

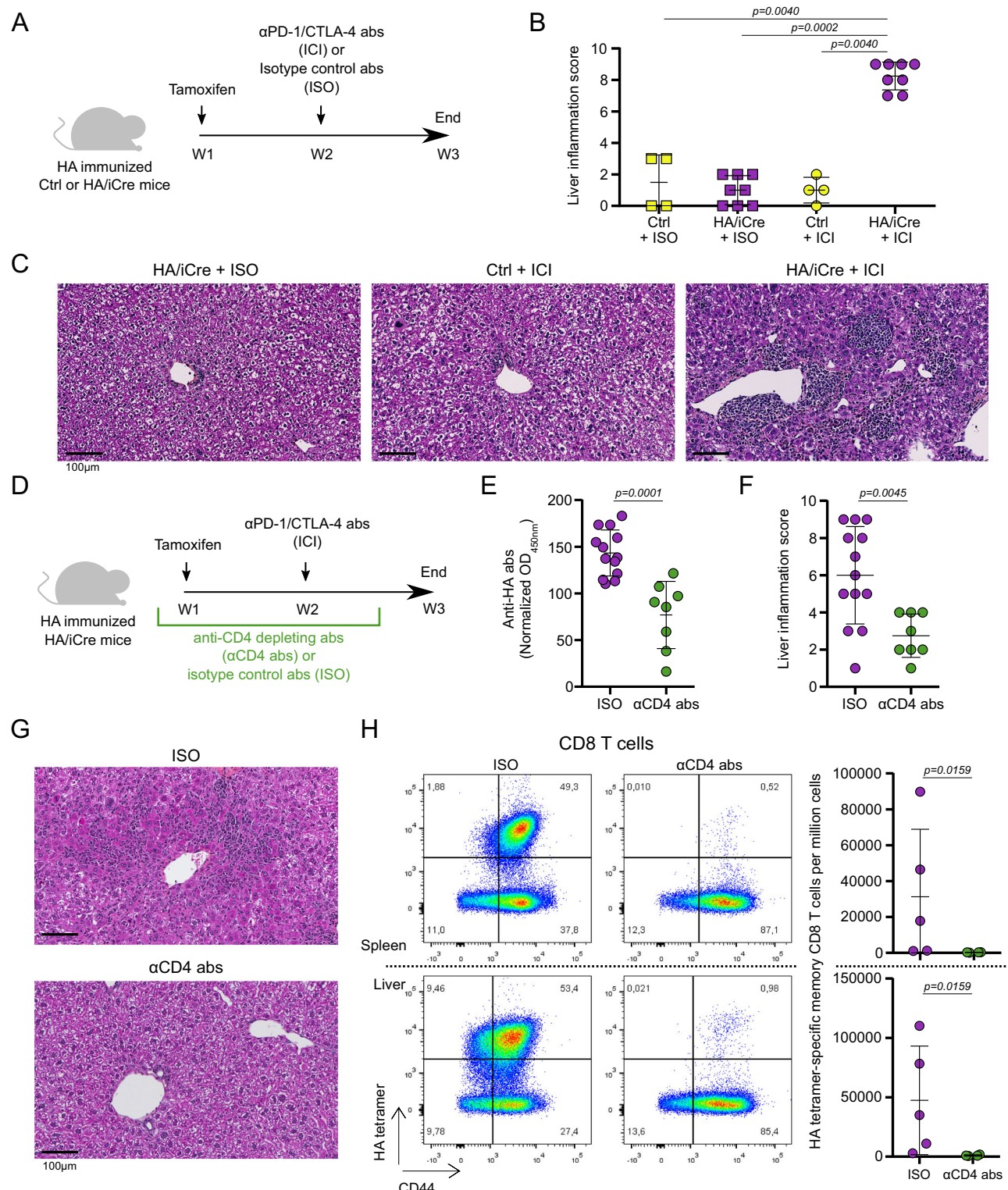

These differences between acute and chronic hepatitis suggest a possible gradual escape from standard immuno-regulatory processes of the liver-self-antigen-specific T cells during AILD. Studies have shown links between CTLA-4 polymorphisms and susceptibility to AIH and PBC[70–75]. However, the functional biological link with the development of AILD is not known. We can hypothesize that a dysregulation of this pathway participates in the gradual temporal escape of liver-antigen-specific CD4 T cell response that mediates the expansion of cytotoxic CD8 T cells. This hypothesis may be supported by the fact that we were able to detect persistent self-antigen-specific CD4 T cell clones for three years, suggesting maintenance of clonal expansion. Therefore, understanding the link between genetic risk and the functional capacity of autoreactive T cells to escape tolerance may open new therapeutic prospects.

## Methods

### Patients

All the patients eligible signed a written informed consent prior to inclusion into a bio-bank of samples of AILD patients (BIO-MAI-FOIE) maintained in Nantes University Hospital which obtained regulatory

**Fig. 7 | Immune checkpoint pathways control liver antigen-specific CD4 T cell responses and mediate hepatic tolerance. A** Experimental design for PD-1 and CTLA-4 blockade (ICI) in HA immunized tamoxifen-treated HA/iCre mice. Injection of isotype control antibodies to ICI was used as control (ISO). **B** Histological liver inflammation scoring analysis of liver tissue sections from immunized tamoxifen-treated control (Ctrl; $n = 8$) and HA/iCre ($n = 16$) mice treated either with isotype control antibodies (+ISO; Ctrl: $n = 4$, HA/iCre: $n = 8$) or anti-PD-1 and anti-CTLA-4 blocking antibodies (+ICI; Ctrl: $n = 4$, HA/iCre: $n = 8$). **C** Representative pictures of paraffin-embedded liver sections stained with HPS coloration from indicated conditions. Black line is used as scale. **D** Experimental design for CD4 depletion (αCD4 abs) in immunized tamoxifen-treated HA/iCre mice receiving PD-1 and CTLA-4 blockade protocol ($n = 13$). Injection of isotype control antibody to αCD4 abs was used as control (ISO; $n = 8$). **E** Analysis of normalized anti-HA antibody rate in serum of mice. **F** Histological liver inflammation scoring analysis of liver tissue sections. **G** Representative pictures of paraffin-embedded liver sections stained with HPS coloration from indicated conditions. Black line is used as scale. **H** Pseudocolor dot plot representation of HA tetramer staining and CD44 expression in CD8 T cells from spleen (top) and liver (bottom) of ISO ($n = 5$) and αCD4 abs ($n = 4$) mice (left). Frequency of HA tetramer-specific memory (CD44$^{high}$) CD8 T cells per million cells (right). Data are presented as mean values ± SD in graphs (**B**, **E**, **F**, and **H**). Two-sided Mann-Whitney test was used for (**B**, **E**, **F**, and **H**). $p$-values are indicated. Source data are provided as a Source Data file.

clearance from the Biomedical Research Ethics Committee (COMITE DE PROTECTION DES PERSONNES OUEST IV-NANTES CPP) and all required French Research Ministries authorizations to collect biological samples (Ministère de la Recherche, ref MESR DC-2017-2987). The Biobank is supported by the HEPATIMGO network promoted since 2017 (RC17_0228) by Nantes University Hospital and is a prospective multi-centric collection managed by the Biological Resource Center of the CHU of Nantes. All data collected were treated confidentially and participants are free to withdraw from the study at any time, without explanation and without prejudice to their future care. It was granted authorization from the CNIL: 2001209v0. All AIH patients included in this study had a simplified diagnostic score superior or equal to 6 according to the simplified scoring system for AIH of the international autoimmune hepatitis group (IAHG)[5,76]. PBC patients included in this study were diagnosed based on cholestasis, increased serum IgM, and presence of anti-mitochondria M2 (anti-M2) antibody[7]. Active AIH (AIHa) patients are untreated patients with new onset AIH patients enrolled at diagnosis prior any treatment initiation as previously described[66], and AIH patients under standard treatment but do not normalize the transaminases (AST and ALT) and/or the serum IgG levels or are under relapsing event. Remission AIH (AIHr) patients are defined biochemically by a normalization of the transaminases and the IgG levels, according to the most recent European clinical practice guidelines on AIH[5]. All NASH (non-Alcoholic SteatoHepatitis) patients had histological evidence of NASH and had dysmetabolic syndromes. No sex analysis was carried out, as AILD are rare disease that predominantly affects women[5,7]. This study was carried out in accordance with the Principles of International Conference on Harmonisation (ICH) Good Clinical Practice (GCP) (as adopted in France) which builds upon the ethical codes contained in the current version of the Declaration of Helsinki, the rules and recommendations of good international (ICH) and French clinical practice (good clinical practice guidelines for biomedical research on medicinal products for human use) and the European regulations and/or national legislation and regulations on clinical trials.

## Peptide re-stimulation assay
10 to $20 \times 10^6$ PBMCs (at a final concentration of $10 \times 10^6$/mL) were stimulated for 3 to 4 h at 37 °C with 1 μg/mL of synthesized peptides spanning all of the Sepsecs, CYP2D6 or PDCE2 sequences (20 amino acids in length with a 12 amino acid overlap; Synpeptide, China) or with 1 μg/mL of PepTivator C. albicans MP65 (130-096-776), Influenza A (H1N1) MP1 (130-097-285) and SARS-CoV-2 Prot_S (SPIKE; 130-126-701) (peptides pools of 15 amino acids length with 11 amino acid overlap, Miltenyi Biotec) in 5% human serum RPMI medium in the presence of 1 μg/ml anti-CD40 (130-094-133, HB14, Miltenyi Biotec). After 4 hrs of specific peptide stimulation, PBMCs were first labeled with PE-conjugated anti-CD154 (30-113-607, 5C8, Miltenyi Biotec), and CD154$^+$ cells were then enriched using anti-PE magnetic beads (130-048-801, Miltenyi Biotec) and magnetic MS column (130-042-201, Miltenyi Biotec). A 1/10th fraction of non-enriched cells was saved before enrichment for frequency determination. Frequency was calculated with the formula $F = n/N$, where $n$ is the number of CD154 positive cells

in the bound fraction after enrichment and N is the total number of CD4$^+$ T cells (calculated as 10× the number of CD4$^+$ T cells in 1/10th non-enriched fraction that was saved for analysis). After enrichment, cells were stained with appropriate antibodies (Supplementary Table 2).

## Tetramer staining
For tetramer staining, 30 million PBMCs in 200 μL of 5% human serum RPMI medium were stained with 20 μg/mL PE-labeled tetramers at room temperature for 100 min. Cells were washed and incubated with anti-PE magnetic beads (Miltenyi Biotec, Germany), and a one-tenth fraction was saved for analysis. The other fraction was passed through a magnetic MS column (Miltenyi Biotec). After enrichment, cells were stained with appropriate antibodies (Supplementary Table 2). Sepsecs$_{187-197}$ HLA-DRB1 03:01 tetramer was generated by the tetramer core at the Benaroya Research Institute (Seattle, USA).

## Flow cytometry and cell sorting
All antibodies used are described in the Supplementary Table 2. Briefly, for surface staining, PBMCs were incubated 20 min with a mix of antibodies and then washed prior analysis or cell sorting on BD FACSCantoII, Cytek Aurora, or BD FACSAriaII. For intracellular staining, we used the Fixation/Permeabilization Solution Kit (554722, BD Cytofix/Cytoperm, BD Biosciences).

## Single cell RNA sequencing and analysis (plate-based assays)
For scRNA-seq analysis of peptide-restimulated CD4 T cells, CD154$^+$ memory CD4 T cells were first sorted on BD FACSAriaII, one cell per well, in 96-well plates containing specific lysis buffer at the CR2TI, Nantes. Plates were immediately frozen for storage at −80 °C, and sent on dry ice to the Genomics core facility of CIML, Marseille, for further generating scRNAseq libraries with the FB5P-seq protocol as described[22,77]. Briefly, mRNA reverse transcription (RT), cDNA 5′-end barcoding, and PCR amplification were performed with a template switching (TS) approach. After amplification, barcoded full-length cDNA from each well were pooled for purification and library preparation. For each plate, an Illumina sequencing library targeting the 5′-end of barcoded cDNA was prepared by a modified transposase-based method incorporating a plate-associated i7 barcode. Resulting libraries had a broad size distribution, resulting in gene template reads covering the 5′-end of transcripts from the 3rd to the 60th percentile of gene body length on average. As a consequence, sequencing reads covered the whole variable and a significant portion of the constant region of the TCRα and TCRβ expressed mRNAs, enabling assembly and reconstitution of TCR repertoire from scRNAseq data. Libraries prepared with the FB5P-seq protocol were sequenced on Illumina NextSeq2000 platform with P2 100-cycle kits, targeting $5 \times 10^5 - 1 \times 10^6$ reads per cell in paired-end single-index mode with the following configuration: Read1 (gene template) 103 cycles, Read i7 (plate barcode) 8 cycles, Read2 (cell barcode and Unique Molecular Identifier) 16 cycles. We then used a custom bioinformatics pipeline to process fastq files and generate single-cell gene expression matrices and TCR sequence files, as described[22,77].

For scRNA-seq of tetramer-stained CD4 T cells, we used an updated version of FB5P-seq (Flash-FB5P-seq) modified to implement advances developed by Hahaut et al. in the Flash-seq protocol[78]. Cells were sorted in 2 μl lysis mix composed of 0.1% Triton X-100 (0.02 μl), 1.2 U/μl recombinant RNAse inhibitor (0.06 μl), 6 mM dNTP mix (0.48 μl), 1.6 mM RT primer (5′-TGCGGTATCTAAAGCGGTGAGTTTTT TTTTTTTTTTTTTTTTTTTTTTTTTT*V*N-3′) (0.36 μl), 0.0125 pg/μl ERCC spike-in mix (0.05 μl), 9 mM dCTP (0.18 μl), 1 M betaine (0.4 μl), 1.2 mM DTT (0.024 μl) and PCR-grade $H_2O$ (0.426 μl). Plates were conserved at −80 °C until sorting, thawed at RT for cell sorting, and immediately refrozen on dry ice then conserved at −80 °C until further processing. Reverse transcription with template switching and cDNA amplification were performed in a single step. Each plate containing cells in lysis mix was thawed on ice, heated at 72 °C for 3 min then immediately placed on ice. To each well containing 2 μl cell lysate, we added 8 μl of an RT-PCR mix composed of 2 U/μl Superscript IV reverse transcriptase (0.1 μl), 0.8 U/μl recombinant RNAse inhibitor (0.2 μl), 1X KAPA HiFi ReadyMix (5 μl), 4.8 mM DTT (0.48 μl), 0.8 M betaine (1.6 μl), 9.2 mM $MgCl_2$ (0.092 μl), 0.1 μM forward PCR primer (5′-AGACGTGTGCTCT TCCGAT*C*T-3′) (0.1 μl), 0.1 μM reverse PCR primer (5′-TGCGGTATC-TAAAGCGGTG*A*G-3′) (0.1 μl), 1.84 μM biotinylated barcoding template switching oligonucleotide (5′-biotin-AGACGTGTGCTCTTCCGA TCTXXXXXXXXXNNNNNNNNNCAGCArGrGrG-3′, where XXXXXXXX are well-specific barcodes as published[77], NNNNNNNN are random nucleotides serving as Unique Molecular Identifiers, and rG are ribo-guanosine RNA bases) (0.184 μl) and PCR-grade $H_2O$ (0.144 μl). Then plates were incubated in a thermal cycler 60 min at 50 °C, 3 min at 98 °C, and 22 cycles of 20 s at 98 °C, 20 s at 67 °C and 6 min at 72 °C. We then pooled 5 μl amplified cDNA from all wells into one tube per plate and performed 0.6X SPRI bead-based purification. Sequencing libraries were prepared from 800 pg cDNA per plate with the Illumina Nextera XT protocol, modified to target sequencing reads to the 5′ end of cDNA, as described[22,77]. Before sequencing, pooled indexed libraries were depleted from ribosomal RNA-derived cDNA molecules by using the SEQoia RiboDepletion Kit (Biorad) according to the manufacturer's instructions. Flash-FB5P-seq libraries were sequenced on an Illumina NextSeq2000 using P3 100 cycle kit and targeting $5 \times 10^5$-$1 \times 10^6$ reads per cell in paired-end single-index mode with the following configuration: Read1 (gene template) 114 cycles, Read i7 (plate barcode) 8 cycles, Read2 (cell barcode and Unique Molecular Identifier) 16 cycles. We then used a custom bioinformatics pipeline to process fastq files and generate single-cell gene expression matrices and TCR sequence files. Briefly, reads were processed with the zUMIs pipeline[79] to generate the gene expression UMI count matrix, and with the TRUST4 pipeline[80] to reconstruct TCR sequences.

Quality control excluded cells with more than 10% mitochondrial UMIs, less than 400 genes, more than 25% ERCC UMIs, less than 0.7 ERCC spike-in quantification accuracy[77], and less than 5% ribosomal protein coding UMIs. Gene expression UMI matrices were analyzed with the *Seurat v4.1.0* package in R v4.1.2 using standard log normalization, 4000 highly variable genes (computed with the *vst* method) from which we removed all TCR coding genes, 40 principal components for computing UMAP, and Louvain clustering at resolution 0.8. Marker genes of unsupervised clusters or metadata-defined clusters were computed with the *FindAllMarkers* function using Wilcoxon test with an adjusted p-value cutoff of 0.01. Gene expression heat maps were generated with the *pheatmap* function. TCR clonotypes were computed and analyzed as described previously[22].

### Single cell RNA sequencing and analysis (10X Genomics protocol)

First, PD-1$^+$CD45RA$^-$CXCR5$^-$ and PD-1$^-$CD45RA$^-$CXCR5$^-$ CD4 T cells were sorted on BD FACSAriaII. Cells were counted, centrifuged (500 × *g*, 5 min, 4 °C) then resuspended at the recommended dilution for a 20000 cell loading per sample onto a Next GEM Chip K (PN-1000286)

and run on a Chromium Single Cell Controller using the Chromium Single Cell 5′ V2 Next GEM single cell kit (PN-1000263) according to the manufacturer's instructions (CG000331, 10X Genomics). Gex and VDJ-TCR libraries were then prepared (PN-1000190, PN-1000252, PN-1000215 also from 10X genomics), checked for quality controls, and sequenced on a S2 flow cell on a Nova-Seq 6000 (Illumina) at the GenoBird platform (IRS-UN, CHU Nantes). Raw reads were analyzed using FastQC for quality controls and were then processed using CellRanger pipeline (v3.1.0 with default parameters). Generated FASTQ files were aligned to the human reference genome GRCh38.

The base calling has been done using Illumina bcl2fastq2 (v2.20). The FASTQs obtained has been analyzed for cell identification and UMIs count using 10X Genomics Cell Ranger 7.0.1 and hg38 reference. Four filtered matrices were obtained for each sample and the script written for their analysis has been written in R (v4.2.2). The matrices have been analyzed using Seurat 4[81] using default parameters. Each matrix has been filtered with the following cut-off: 500 <Feature-s_RNA < 4000; percent.mt <5 and cells having two distinct TRA and TRB identified have been removed for the rest of the analysis. Then each matrix was normalized using the SCTransform method with a regression on the percentage of ribosomal gene count. The four matrices has been merged and integrated using harmony package[82]. The cell clusters have been identified using harmony for reduction. The markers for each clusters have been identified with a min.pct=0.25 (genes detected in a minimum fraction of min.pct cells) and a threshold of logFoldChange of 0.25. The markers retained for each cluster where those having a *p*-value adjusted <0.01. The annotation of clusters has been done using azimuth package[81]. The module score has been calculated for each cluster using a predefined list of genes for "Auto-reactive", "SARS-CoV-2", "H1N1" and "C.ALB". The top five percent of cells base on this module score have been highlighted on the UMAP. The clusters have been then compared based on their module score for each gene list using Kruskal Wallis test followed by a paired Wilcoxon rank test.

### Mini-bulk-RNA sequencing and analysis

For mini-bulk RNA-seq of in vitro stimulated CD4 T cells, we used a slightly modified version of the Flash-FB5P-seq protocol described above. First, CD4 T cell subsets were sorted on BD FACSAriaII. 500 to 2000 cells per well were stimulated in presence of 2.5 μg/ml anti-CD3 and 1 μg/ml anti-CD28. After 24 h, cells were washed gently in PBS and resuspended in 4 μL of specific lysis buffer with the composition described above except for ERCC spike-in mix which we used 10 times more concentrated (0.125 pg/μl). We performed RT-PCR with 16 μl of the RT-PCR mix described above, using 16 cycles of PCR for cDNA amplification (instead of 22 cycles for single-cell assay). Sequencing libraries were prepared and sequenced as described above for Flash-FB5P-seq. Sequencing reads were processed with the zUMIs pipeline[79] to generate gene expression UMI count matrices. Mini-bulk gene expression UMI count matrices were pooled across plates and further processed with the *DEseq2* package for normalization (*vst* method) and computation of differentially expressed genes between non-stimulated and TCR-stimulated cells from each sorted phenotype. The mean of normalized expression values for manually selected significant DE genes was used to construct the heatmaps in Fig. 5F.

### Bulk TCRβ sequencing

$100 \times 10^3$ PD-1$^+$CD45RA$^-$CXCR5$^-$ and PD-1$^-$CD45RA$^-$CXCR5$^-$ CD4 T cell subsets were sorted on a BD FACSAriaII. Genomic DNA from T cells and frozen liver biopsies was extracted with the NucleoSpin Blood kit (740951.50, Macherey-Nagel) and the rearranged V, D, and J gene segments of the human TRB (TRBV, TRBD, and TRBJ) loci were amplified in a multiplex polymerase chain reactions (PCR) using the BIOMED2-TRB-B primer pool and 250–500 ng template DNA. Addition of Illumina-compatible adapters and seven nt barcodes were added in

two consecutive PCR reactions, purified with the NucleoSpin Gel and PCR Clean-up kit (Macherey-Nagel, Düren, Germany), quantified using the Qubit platform (QIAGEN, Hilden, Germany), and pooled to a final concentration of 8 nM. Pools were quality-controlled on an Agilent 2100 Bionanalyer (Agilent Technologies, Böblingen, Germany). Sequencing and demultiplexing were performed on an Illumina MiSeq sequencer (600-cycle single-indexed, paired-end run, V3-chemistry), raw data was aligned using the MiXCR framework, and the implemented default TRB reference library. Non-productive reads and sequences with less than two read counts were discarded. Each unique complementarity-determining region 3 (CDR3) nucleotide sequence was considered a clone. All analyses and data plotting were performed using RStudio (version 1.1.456) and the tcR, ade4, and tidyverse packages[83,84].

## Unsupervised flow cytometry analysis

Flow cytometry data from patients were analyzed using Omiq software (Dotmatics, USA). Subsampling of viable lymphoid CD45RA- CD4+ CD3+ cells was performed ($20 \times 10^3$ per patient). Clustering was performed using FlowSOM to generate 150 clusters. 22 Metaclusters were generated by running consensus metaclustering. Wilcoxon test was used to identify significant variation between two groups of patients.

## Spatial multi-phenotyping and analysis (PhenoCycler instrument)

The PhenoCycler instrument (Akoya Biosciences, USA) performs iterative annealing and removal of fluorophore-conjugated oligo probes to primary antibody-conjugated complementary DNA barcodes. Antibody panel was constructed using 11 ready-to-use commercially available PhenoCycler antibodies (Akoya Biosciences) and one unavailable antibody has been custom conjugated. Detailed 12-plex PhenoCycler information is provided in Supplementary Table 2. Anti-CXCR5 antibody has been custom conjugated with a unique PhenoCycler oligonucleotide tag using the antibody conjugation kit of Akoya Biosciences following their instructions (PhenoCycler conjugation kit, 7000009, Akoya Biosciences)

FFPE human liver was sectioned to a thickness of 5 μm and directly adhered onto poly-L-lysine (Sigma) coated 22 × 22 mm coverslips (7000005, Akoya Biosciences). Tissue coverslips were stored at 4 °C until staining. The sample was incubated for 20 min at 55 °C on a hot plate and tissue was cooled down before staining. Tissue section was deparaffinized and rehydrated by immersing the coverslip through the following solution for 5 min each: twice in OTTIX (X0076, Diapath), twice in 100% Ethanol (VWR Chemicals), once in 90%, 70%, 50%, 30% Ethanol and twice in ddH2O. Antigen retrieval was performed in a hot water bath at 100 °C in a 1X Citrate Buffer (Sigma) for 30 min. After cooling at room temperature, tissue section was briefly washed twice in ddH2O for 2 min. Following hydration (Hydration buffer, 240196, Akoya Biosciences) step, tissue section was blocked in Staining buffer (PhenoCycler staining kit, 7000008, Akoya Biosciences) at room temperature for 20 min and stained with the 12 plex PhenoCycler antibody panel in a humidity chamber 3 h at room temperature. To ensure that the antibodies remain attached to the antigen during the multicycles PhenoCycler imaging, protocol post-staining protocol has 3 fixing steps. After incubation with antibodies, tissue section was washed twice and post-fixed with 1.6% PFA (Sigma) Storage buffer (232107, Akoya Biosciences) for 10 min at room temperature. Tissue sections were briefly washed three times in 1X PBS buffer (VWR Life Science) and incubated in a cold methanol solution (Sigma) for 5 min. Following washing, tissue section was fixed in final fixative solution (Akoya Biosciences) for 20 min at room temperature. Finally, tissue section was washed three times in 1X PBS buffer and was stored in Storage Buffer at 4 °C before PhenoCycler imaging.

PhenoCycler imaging was performed with an Axio Observer (Zeiss) inverted microscope equipped with camera ORCA Flash 4.0 LT (Hamamatsu). The microscope is coupled to the Colibri 7 LED light source (Zeiss) and fluorescence was detected using the monoband filter Set 112 HE LED (Zeiss). The PhenoCycler experimental run was managed by the instrument controller software (v1.30.0.12, Akoya Biosciences) integrating with the Zeiss microscope. Nuclear DAPI (Nuclear Stain, 7000003, Akoya Bioscience) staining was used to design manually tiled regions of interest at the 5x magnification (N-Achro 5x/0.15 M27, Zeiss). Automated multiplex imaging was performed using a Plan Apo 20x/0.8 M27 Air objective (Zeiss) with a 325 × 325 nm pixel size using software autofocus repeated every tile before acquiring an 11 plane-z-stack with a z-spacing of 1.5 μm. Data are acquired as 3D stacks by cycle of 4 markers (including a recurrent nuclear DAPI staining). After acquisition, raw files were exported using the CODEX Instrument Manager (CIM, Akoya Biosciences). The data are processed by computing the more in focus image from the stack to take into account the potential tissue unflatness, all cycles are registered together based on the DAPI staining, and the fluorescence signal is corrected by background fluorescence based on blank cycles and normalized in intensity[85]. Cell segmentation was done by using Cellpose algorithm[86]. After segmentation, measurement data are transformed in FCS files and analyzed with the CODEX MAV software.

## Mice and treatments

All mice were housed at the UTE IRS-UN animal facilities (Nantes, FRANCE) in specific pathogen-free conditions. Mice were housed at a maximum of five mice per individually ventilated cage in 12 h:12 h light/dark cycle and under standard conditions of temperature (21–24 °C) and humidity (40–60%). Mice were fed ad libitum with continuous access to tap water. Procedures were approved by the regional ethical committee for animal care (Comité d'éthique de l'expérimentation animale des Pays de la Loire (CEEA-Pdl)) and by the Ministère chargé de l'enseignement supérieur et de la recherche (agreements APAFIS #2054, #28582 and #43529). All experiments were performed in accordance with relevant guidelines and regulations. Heterozygous transthyretin (TTR)-inducible Cre (iCre) mice, originally on a C57Bl6 background[87], were back-crossed on a Balb/c background for at least 10 generations (TAAM, CDTA CNRS Orléans, FRANCE). They were cross-bred with homozygous Rosa26 hemagglutinin (HA) floxed mice (Rosa26tm(HA)1Libl, kindly provided by R. Liblau, Toulouse, France[88]), resulting in heterozygous Rosa26 HA floxed (Ctrl) mice and Rosa26 HA floxed TTR-inducible Cre (HA/iCre) mice. Mice have genotyped for Cre-coding gene at 3 weeks old thanks to small tail biopsies using Cre primers listed in Supplementary Table 3. Male and female eight to twelve-week-old mice were used for each experiment. Experimental and control animals were co-housed for each experiment. At the end of experiments, mice were anaesthetized and euthanized by cervical dislocation.

Induction of HA expression by hepatocytes was performed by feeding mice with tamoxifen dry food (0.5 g/kg tamoxifen + 5% saccharose; Safe, FRANCE) for 14 days in free access or by injecting intraperitonially (i.p) 1 mg of tamoxifen (T5648, Sigma) diluted in corn oil (C8267, Sigma) three times in two weeks span. Peripheral immunization against HA was performed with two different materials: one intramuscular (i.m) injection of $1,5.10^9$ infectious particle of adenoviral CAG Cre vectors (AdCre, produced by INSERM UMR 1089 CPV facility, Nantes, FRANCE) in *tibialis anterior* of one posterior leg or i.m injection of 50 μg of plasmid CMV HA vector in *tibialis anterior* of both posterior legs, twice three weeks apart. For plasmid CMV HA vector, HA sequence (Influenza A virus (A/Puerto Rico/8/1934(H1N1), Gene ID: 956529) was synthesized (GenScript, NETHERLANDS) and cloned in pCMVβ vector (Clontech Laboratories) in place of β-galactosidase gene.

Immune checkpoint blockade protocol was performed by intravenous (i.v) injection of 200 μg of anti-mouse PD-1 antibody and 200 μg of anti-mouse CTLA-4 antibody, using purified Rat IgG2a,κ isotype control antibody and polyclonal Syrian hamster IgG as

controls, three times in one week span. Depletion of CD4$^+$ cells was performed by i.p injection of 500 µg of anti-mouse CD4 antibody, using purified Rat IgG2b,κ isotype control antibody, every two days during 2 weeks. All antibodies used during in vivo experiments are described in the Supplementary Table 2.

## RNA extraction, reverse transcription, and quantitative PCR

Total RNA was extracted from organ tissue using TRIzol reagent (15596026, ThermoFisher Scientific) and purified with RNeasy Mini Kit (74106, Qiagen) according to the manufacturer protocol. Reverse transcription was performed using 2 µg of total RNA incubated at 70 °C for 10 min with poly-dT24 20 µg/mL (Eurofins Genomics), 8 mM DTT (18057018, ThermoFisher Scientific), and 20 mM of each dNTP (10297018, ThermoFisher Scientific). After a brief 5 min incubation at 4 °C, first strand buffer 5X (Y00146, Invitrogen), 200U of M-MLV reverse transcriptase (18057018, ThermoFisher Scientific), and 40U of RNAse OUT inhibitor (10777019, Invitrogen) were added and incubated at 37 °C for 1 h followed by 15 min at 70 °C. Real-time quantitative PCR was performed using the ViiA 7 Real-Time PCR System and Power SYBR Green PCR Master Mix (4368708, ThermoFisher Scientific). Primers used for HA, Cre, and ACTB relative mRNA expression analysis are listed in the Supplementary Table 3. Relative HA mRNA expression was determined with $2^{-\Delta\Delta CT}$ calculation, taking ACTB as reference gene and cDNA from one conserved mouse that received i.v injection of AdCre (AdCre i.v) as positive control. Relative Cre mRNA expression was determined with $2^{-\Delta CT}$ calculation, taking ACTB as reference gene.

## ELISA test

Blood sample clotted for 1 h at room temperature followed by centrifugation at $3000 \times g$ for 10 min and serum was harvested and stored at −80 °C. For detection of anti-HA antibodies, wells were coated with 1 µg/mL of HA protein (11684-V08H, SinoBiological) diluted in coating buffer ($Na_2CO_3$ 0.05 M; $NaHCO_3$ 0.05 M; pH 9.2) and plate was incubated at 4 °C overnight. Wells were washed and saturated with dilution buffer (PBS 1X; Tween 20 0.05%; BSA 1%) for 2 h at 37 °C. Wells were washed, serum samples were diluted by 500, 1000, and 5000 and added to wells in duplicate, and plate was incubated at 37 °C for 2 h. Wells were washed and 0,4 ng/mL of detection antibody (peroxidase-conjugated goat anti-mouse IgG+IgM (H + L) polyclonal antibody, 115-036-068, Jackson ImmunoResearch) was added and plate was incubated at 37 °C for 1 h. Finally, wells were washed and 50 µL of TMB Substrate Reagent (555214, BD Biosciences) was added for revelation step. Reaction was stopped 1min30sec after with $H_2SO_4$ 0.5 M. Optical density (OD) at 450 nm and 620 nm were determined using a Spark 10 M Infinite M200 Pro plate reader (TECAN). Analysis was performed by subtracting $OD_{620nm}$ to $OD_{450nm}$ value, calculating the mean of each duplicate, and subtracting the mean of blank wells to the mean of samples. Then, OD values of each dilution of sample were normalized to OD values of each corresponding dilution of positive control serum sample from one conserved AdCre i.v mouse. Finally, for each sample, mean of normalized $OD_{450nm}$ for each dilution was calculated and analyzed.

## Cell preparation, tetramer enrichment, and staining, and flow cytometry

Splenocytes were isolated by mechanical dissociation of spleen in red blood cell lysis buffer ($NH_4Cl$ 155 mM; $KHCO_3$ 10 mM; EDTA 1 mM; Tris 17 mM per 1 L of sterilized water). Liver non-parenchymal cells (NPCs) were isolated as previously described[89] from liver sample perfused with HBSS 1X. Livers were cut and digested with collagenase IV (C5138, Sigma) at 37 °C for 15 min, followed by crushing step on 70 µm filter. After the first centrifugation step (100 g, 5 min at 4 °C), the supernatant was harvested and underwent a second centrifugation step ($400 \times g$, 12 min at 4 °C). Then pellets were kept and liver NPCs were

enriched by 40%:80% Percoll (GE17-0891-01, Sigma) density gradient centrifugation and red blood cells lysis. $20.10^6$ splenocytes and $3.10^6$ liver NPCs were stained with both I-A$^d$-HA peptide (HNTNGVTAACSHE) and I-E$^d$-HA peptide (SFERFEIFPKE) PE-labeled tetramers (NIH Tetramer Core Facility; Atlanta, USA) at room temperature during 1 h. Cells were washed and stained with magnetic anti-PE microbeads (130-048-801, Miltenyi) at 4 °C during 15 min. Cells were washed and enriched using magnetic MS columns (130-042-201, Miltenyi). Positive fraction was kept for HA-specific CD4$^+$ T cell detection. Negative fraction was stained with H-2K$^d$-HA peptide (IYSTVASSL) PE-labeled tetramer (NIH Tetramer Core Facility; Atlanta, USA) at room temperature during 1 h for HA-specific CD8$^+$ T cell detection. All cells were washed and viability staining (LIVE/DEAD Fixable Aqua Dead Cell Stain kit, dilution 1:1000, L34957, ThermoFisher Scientific) was performed at 4 °C during 15 min, protected from light. Cells were washed and extracellular staining was performed at 4 °C during 20 min protected from light, using fluorochrome-coupled anti-mouse antibodies (Supplementary Table 2). Fluorescence was measured on BD FACSCantoII or BD FACSAriaII (BD Biosciences; Mountain View, USA). HA-specific memory CD4 and CD8 T lymphocytes were defined as: LIVE/DEAD$^-$ CD19$^-$ CD4$^+$ (or CD8$^+$) CD44$^{high}$ tetramer$^+$ cells, as described in Supplementary Fig. 20. FlowJo software (TreeStar Inc) was used to analyze flow cytometry data.

## 3′-end bulk RNA sequencing

50 up to 200 HA-specific memory CD4$^+$ T cells were FACS sorted. RNA extraction, reverse transcription, and PCR steps were directly performed after sorting according to the SMART-Seq v4 Ultra Low Input RNA Kit protocol (Takara Bio, USA). Amplified complementary DNA was transferred to the Benaroya Research Institute (Seattle, USA) to perform a 3′-end bulk RNA sequencing. To normalize, trimmed-mean of M values (TMM) normalization on the raw gene counts was performed. Additionally, because many genes are not expressed in any of the samples and are therefore uninformative, a filtering step was performed requiring that genes be expressed with at least 1 count per million total reads in at least 10% of the total number of libraries. For the differential gene expression analysis, we used the following threshold: fold change (FC) > 2 and adjusted $p$-value < 0.1.

## Histological analysis

Liver lobes were fixed in formol 4% (11699404, VWR) for 24 h at room temperature. Dehydration in differential absolute ethanol baths, embedding in paraffin, and staining of paraffin-embedded sections (3 µm) with hematoxylin-phloxin-saffron (HPS) coloration were performed by the IBISA MicroPICell facility (Biogenouest; Nantes, FRANCE). Slides were scanned and observed using NanoZoomer (Hamamatsu) and NDP Scan software. Histological analysis was performed thanks to a liver inflammation scoring method based on established hepatitis grading[90], simplified to portal inflammation score (0 to 4), lobular inflammation score (0 to 4), and presence of interface hepatitis (0 to 1), giving a minimum score of 0 and maximum score of 9. Liver inflammation scoring was performed blindly.

## Western blot

Total proteins were extracted from liver samples via RIPA buffer treatment. 25 µg of protein were denatured at 95 °C for 5 min in Laemmli Sample Buffer (161747, Bio-Rad) with DTT 0.1 M (10197777001, Sigma). Preparation was separated by SDS-PAGE on Mini-PROTEAN TGX Precast Protein Gels (4561036, Bio-Rad) in migration buffer (Tris 15 g/L; Glycine 72 g/L; SDS 10 g/L) and transferred onto a PVDF membrane with the Trans-Blot Turbo Transfer System (Bio-Rad). The membrane was stained with Ponceau S solution (P7170, Sigma) and was cut horizontally above the 50 kDa molecular weight ladder. Both parts of the membrane were blocked for 2 h with a blocking solution (TBS; Tween 20 0.1%; Skim milk 5%), followed by

overnight incubation at 4 °C with anti-HA antibody (rabbit polyclonal antibody; 11684-T62, Sinobiological) for the upper membrane and anti-mouse ACTB antibody (mouse monoclonal antibody; 3700, Cell Signaling) for the lower membrane, as primary antibodies. The membrane was washed and incubated 1 h at room temperature with peroxidase-conjugated donkey anti-rabbit IgG (H + L) antibody (E-AB-1080-120, Clinisciences) and peroxidase-conjugated goat anti-mouse IgG+IgM (H + L) antibody (115-036-068, Jackson ImmunoResearch). Revelation was performed by Electrochemioluminescence Super Signal West Pico (34577, ThermoFisher Scientific) according to the manufacturer's instructions. Imaging and analysis of western blots were performed on the ChemiDoc MP Imaging System (Bio-Rad). Total protein extract from liver sample of one conserved AdCre i.v mouse and one wild-type Ctrl mouse was used as positive and negative controls respectively.

**ELISpot test.** The first day, plates were coated with 5 µg/mL of BD NA/LE Purified anti-mouse IFN-γ capture antibody (551881, BD Biosciences) diluted in sterile PBS and incubated overnight at 4 °C. The second day, wells were washed and blocked with RPMI medium (supplemented with 10% FBS 1% Penicillin/Streptomycin, 1% L-Glutamine) for 2 h at room temperature. After the washing step, 100 µL of stimuli were added to plates (5 ng/mL PMA (P8159, Sigma) + 500 ng/mL Ionomycin (I9657, Sigma) was used as positive control; 5 µg/mL HNTNGVTAACSHE and 5 µg/mL SFERFEIFPKE peptides (Sigma) from HA protein were used to stimulate HA-specific CD4 T cells; 5 µg/mL IYSTVASSL peptide (Synpeptide) from HA protein was used to stimulate HA-specific CD8 T cells) and $200.10^3$ cells from mouse spleen or liver were added and plate was incubated at 37 °C overnight in 5% CO2 humidified incubator. The third day, wells were washed and 2 µg/mL of Biotinylated anti-mouse IFN-γ detection antibody (551881, BD Biosciences) diluted in PBS 10% FBS was added and plates were incubated 2 h at room temperature. After washing steps, HRP Streptavidin (557630, BD Biosciences) diluted by 100 in PBS 10% FBS was added to wells and plates were incubated for 1 h at room temperature. Finally, revelation step was performed following AEC Substrate Set protocol (551951, BD Biosciences) and reaction was stopped 30 min after with deionized water and wells were washed several times. Plates were air-dried at room temperature and were red on IRIS 2 reader (Mabtech). Each condition was performed in triplicate. Analysis was performed by converting median of spots counted in triplicates to number of IFNγ-secreting cells per million cells.

### Statistical analysis
Statistical comparisons were performed using GraphPad Prism software V.5 (GraphPad Software, La Jolla, CA, USA) and FaDA[91] (https://shiny-bird.univ-nantes.fr/app/Fada). $P$-value $< 0.05$ after adjustment were considered significant.

### Reporting summary
Further information on research design is available in the Nature Portfolio Reporting Summary linked to this article.

## Data availability
All data are included in the Supplementary Information or available from the authors, as are unique reagents used in this Article. The raw numbers for charts and graphs are available in the Source Data file whenever possible. Source data are provided with this paper. The FCS data (Fig. 5) are available under restricted access due to unpublished and ongoing analysis, access can be obtained upon request within a reasonable timeframe after publication. Single-cell RNA sequencing raw data from the different experiments analyzed in this article have been deposited in the NCBI GEO repository under accession numbers GSE270739, GSE269661, and GSE269525. Processed and annotated single-cell RNA-sequencing datasets analyzed in this article are available in the following repository: https://zenodo.org/records/14516943. Source data are provided with this paper.

## Code availability
Scripts used for analyzing the datasets and producing the figures of this article are available in the following repository: https://github.com/MilpiedLab/Autoreactive-CD4-T-cells-in-liver-disease.

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

## Acknowledgements

We thank the biological resource center for biobanking (CHU Nantes, Hôtel Dieu, Center de ressources biologiques (CRB), Nantes, F-44093, France (BRIF: BB-0033-00040)). We thank all the members of the HEPA-TIMGO network. We thank Dr. Sarah HABES, Dr. Maëva SALIMON, and Dr. Annie LIM for the inclusion of new AIH patients. We acknowledge the MicroPICell core facility (SFR Bonamy, BioCore, Inserm UMS 016, CNRS UAR 3556, Nantes, France), member of the Scientific Interest Group (GIS) Biogenouest, IBISA, and the national infrastructure France-Bioimaging supported by the French national research agency (ANR-10-INBS-04). We thank TaRGeT laboratory and the GTI core (Nantes Université, CHU Nantes, INSERM, TaRGeT, F-44000 Nantes, France) for the help in ELISpot plate reading. We thank the BIRD Core facility of the SFR Santé F. Bonamy (Université de Nantes, Inserm UMSO16, CNRS UMS3556). We thank the Genomics core facility and the Computational Biology, Biostatistics and Modeling (CB2M) hub of CIML. Center de Calcul Intensif d'Aix-Marseille is acknowledged for granting access to its high-performance computing resources. Supported by the Agence Nationale de la Recherche (ANR-19-CE17-0024), the patient association Association pour la Lutte contre les maladies inflammatoires du foie et des voies biliaires (ALBI), the Fondation Maladies Rares (including the High throughput sequencing and rare dis-eases program), the Région pays de la Loire, the LabEx IGO program (n° ANR-11-LABX-0016) funded by the Investment into the Future French Government program managed by the Agence Nationale de la Recherche (ANR). This work was supported by institutional grants from INSERM, CNRS, and Aix-Marseille University to the CIML.

## Author contributions

A. C. and T. G. performed the experiments, analyzed the data, and wrote the manuscript; C. D. analyzed the data; L. G. and S. A. performed the experiments, P-J. G. performed the experiments and analyzed the data; M. B. analyzed the data; R. D. analyzed the data; C. S. performed the experiments and analyzed the data; A. D. performed the experiments; P. P-G. supervised data analysis analyzed the data, and wrote the manu-script; C. C. managed human AILD samples collection and patient data-base; L. B. and J-P. J. performed experiments; C. F. performed experiments; A. I., M. K., E. B-J., L. E., A. L., C. S., M. S., and F. T. provided human AILD samples and critical insight in AILD pathology; F. V. managed human AILD samples collection; L. B. performed the experiments; S. B. provided critical insight the study design, W. W. K. provided the MHC class II tetramer technology; J-F. M. provided human liver AILD biopsies and critical insight in histology analysis; A. W. L. provided human AILD samples and critical insight in AILD pathology; J. P. supervised data analysis; M. Binder supervised data analysis and critical insight TCR sequencing; J. G. provided human AILD samples, critical insight in AILD pathology and direction in the study design; S. C. designed the study, supervised data analysis and wrote the manuscript; P. M. designed the study, supervised data analysis and wrote the manuscript, A. R. designed the study, super-vised data analysis, performed the experiments, analyzed the data and wrote the manuscript. All authors reviewed and approved the manuscript.

## Competing interests

The authors declare no competing interest.

## Additional information

[1]Nantes Université, CHU Nantes, INSERM, Center for Research in Transplantation and Translational Immunology, UMR 1064, Nantes, France. [2]Aix Marseille Université, CNRS, INSERM, Centre d'Immunologie de Marseille-Luminy, CIML, Marseille, France. [3]Laboratory of Translational Immuno-Oncology, Department of Biomedicine, University and University Hospital Basel, Division of Oncology, University Hospital Basel, Basel, Switzerland. [4]Nantes Université, CHU Nantes, CNRS, Inserm, BioCore, US16, SFR Bonamy, Nantes, France. [5]Centre d'Investigation Clinique IMAD, CHU Nantes, Nantes, France. [6]Service Hepato-gastro-entérologie et Assistance Nutritionnelle, CHU Nantes, Nantes, France. [7]Institut des Maladies de l'Appareil Digestif, IMAD, CHU Nantes, Nantes, France. [8]CHU Rennes, Service des maladies du foie, Université Rennes, INSERM, INRAE, Institut NUMECAN, Rennes, France. [9]CHRU Tours, Service Hépato-Gastroentérologie, Tours, France. [10]CHU Angers, Service Hépato-Gastroentérologie et Oncologie Digestive, Université d'Angers, Laboratoire HIFIH, UPRES EA3859, SFR 4208, Angers, France. [11]CHU Poitiers, Service Hépato-Gastroentérologie, Poitiers, France. [12]CHD Vendée-La Roche sur Yon, Service Hépato-Gastroentérologie, F- 85000, la Roche sur Yon, France. [13]CHU Brest, Service Hépato-Gastroentérologie, Brest, France. [14]Center for Translational Immunology, Benaroya Research Institute, Seattle, WA, USA. [15]Service Anatomie et Cytologie Pathologiques, CHU Nantes, Nantes, France. [16]First Department of Medicine, University Medical Center Hamburg-Eppendorf, Hamburg, Germany. [17]These authors contributed equally: Anaïs Cardon, Thomas Guinebretière. [18]These authors jointly supervised this work: Sophie Conchon, Pierre Milpied, Amédée Renand. ✉e-mail: sophie.conchon@univ-nantes.fr; milpied@ciml.univ-mrs.fr; amedee.renand@univ-nantes.fr

