## [Transparent Peer Review file · Nature Communications]

Single cell profiling of circulating autoreactive CD4 T cells from patients with autoimmune liver diseases suggests tissue imprinting

Corresponding Author: Dr Amedee Renand

Version 0:

Reviewer comments:

Reviewer #1

(Remarks to the Author)

Renand and colleagues provide an impressive, multi-layered analysis of antigen-reactive CD4 T cells in a small cohort of patients with autoimmune liver disease. The work extends their prior work (Renand et al, 2020), now using a combination of single cell RNAseq, bulk RNAseq, and TCR analyses of blood CD4 T cells from AILD patients to define the phenotypes, transcriptomes, and TCR repertoire of antigen-reactive T cells. The stepwise analysis of sorted antigen-reactive cells, then both scRNA-seq and bulk RNA-seq of sorted PD1+ cells, which are enriched for Ag-reactive cells, is valuable and informative. The analysis of paired blood and tissue samples, utilizing bulk TCR analyses of liver tissue, provides a valuable connection to the tissue. The analyses are generally internally consistent, supporting the robustness of the overall message and the similarity of Ag-reactive T cells in AILD to Tph cells. The work extends further with a murine model using induced hepatocyte-specific induced HA expression; analysis of the activated T cells in this model reproduce the phenotype of inhibitory receptor expression on Ag-specific T cells. The authors further show a role for these inhibitory receptors with exacerbation of liver disease following checkpoint blockade. In total, this broad analysis involving multiple complementary analyses thoughtfully stitched together provides an excellent demonstration of building a case for likely pathogenic antigen-specific T cells in AILD.

Major comments:

- 1) The Tph-like features of the Ag-reactive cells are observed in several analyses, yet it is hard to connect these features to the scRNAseq data in Figure 3. Is there a cluster that demonstrates Tph-like features such as IL21 or CXCL13 expression? Does cluster2 show an enrichment in Tfh or Tph-like signatures? Similarly, what would a plot of a 'Tfh' or 'Tph' signature look like if plotted in Fig4D as done for CTL, GZMK, HLADR signatures?
- 2) The signature of autoreactivity in Figure 3 highlights both the HLA+ cluster and also the Treg cluster. Do the Ag-reactive T cells show other features suggestive of Tregs? Can the authors provide some insight into this transcriptomic signal?
- 3) Is there TCR clonal sharing across the clusters in Fig 3B (i.e. clusters 1,2,3)? This could be particularly relevant for the T cell clones that match TCRs in liver. Such clonal sharing might provide further insight into the phenotypic diversity of the expanded clones, and perhaps hint at ontogeny of the phenotype most strongly associated with autoreactivity.
- 4) A conceptual point: The authors posit that the autoreactivity-associated phenotype is imprinted in the liver through chronic antigen stimulation. The murine model demonstrating differences between splenic and liver T cells supports this idea. However, there remains uncertainty about whether it is the chronic antigen stimulation, or the liver environment, or both, that are required to induce this phenotype. The fact that a similar phenotype emerges across multiple autoimmune diseases targeting diverse organs and in tumors (e.g. synovium, duodenum, breast tissue) suggests that the PD1+ HLADR+ TIGIT+ phenotype does not require signals from the liver specifically. Is it possible that non-lymphoid tissues share some set of features that promote acquisition of this phenotype? Or is chronic Ag stimulation the major driver, and in the murine model, T cells from liver have been more persistently exposed to antigen than those in spleen? Do the authors have any insight into this?

Reviewer #2

(Remarks to the Author)

In this manuscript, Cardon et al. demonstrated that intrahepatic and peripheral self-antigen specific CD4 T cells showed immuno-exhausted transcriptional phenotype in patients with autoimmune hepatitis, which is distinct from foreign antigen-specific CD4 T cells. They also showed that immune checkpoint molecules controlled the response of antigen-specific CD4 T cells responsible for liver damage. Overall, the data are presented well and the results are potentially interesting, however, there are concerns to be addressed.

Major concerns

1. Information on the phenotypic and transcriptional profiling of intrahepatic and peripheral self-antigen specific CD4 T cells is still insufficient. Recent reports have shown that tissue resident memory T cells (Trm), a relatively new cell subset, are involved in the pathogenesis of autoimmune hepatitis (You et al. Hepatology, 2021) and MASH. The authors need to address this with specific markers of Trm.
2. It is unclear how these antigen-specific CD4 T cells emerge in the liver and acquire the immune-exhausted phenotype after local antigen reactivity in the course of autoimmune hepatitis. What is the origin of these cells?
3. In Figure 7A, the authors demonstrated that ICI treatment exacerbated liver inflammation, but it is unclear whether this phenomenon is due to recovery of antigen-specific CD4 T cell function. To clarify this point, information on the number of HA tet+ CD4 T cells and their function (cytokine-producing capacity, cytotoxic activity, etc.) before and after ICI treatment should be provided.
4. Related to 3, it is also important to know whether patient-derived peripheral and intrahepatic antigen-specific CD4 T cells can be similarly restored to function by immune checkpoints inhibition.

Reviewer #3

(Remarks to the Author)

Cardon & Guinebretiere et al

The immune profile of circulating autoreactive CD4 T cells is imprinted through tissue activation during autoimmune liver diseases

In this study presented by Cardon & Guinebretiere et al the authors present transcriptional and TCR clonotyping data on circulating Ag-specific CD4 T cells in the context of autoimmune liver disease revealing an immuno-exhausted phenotype using a cohort of human samples and a mouse model of AILD. The authors propose a profile imprinted on peripheral CD4 T cells that is linked to clonal expansion in liver.

While this reviewer understands the limitations associated with sampling the liver, it is hard to ignore the increasing body of literature surrounding tissue compartmentalisation, and even the concept of ex-TRMs. This needs to be given more due consideration throughout the manuscript. For example, how do the circulating Ag-specific T cells compare to those in liver transcriptionally? Is there evidence of a tissue-residency profile in these cells? Of tissue priming/re-activation? What transcriptional profile(s) has/ve been altered or down-regulated to enable T cells primed/activated in liver to leave, and re-enter the circulation? Or perhaps the evidence suggests these circulating Ag-specific T cells are a totally distinct subset/phenotype of T cells that have not/will not take up residence in the AILD liver?

Previous work by the group has shown detection of SLA-specific CD4 T cells in blood of those with AIH, here in this study the group extended these analyses to include other specificities in an extended? Further? cohort of AIH and those with different autoimmune conditions (e.g. PBC). For these analyses Ag-specific T cells have been detected using the AIM assay popularised during the recent wave of COVID-19 research (references required to original assay). However, the quantification and presentation of the data is unusual/unsuitable. Post stimulation the others have stained for CD154-PE, enriched for PE-specific T cells (n) and then presented 'responding CD1554 cells' as a fraction of the PE-negative, non-enriched pool (N). Firstly the denominator is wrong, this is not the total CD4 T cell number for the sample, as CD4+ T cells have been removed during enrichment and so the calculation is wrong. But secondly, why are the authors not showing conventional ICS detection or multimer staining. If using AIM, then the data should be presented as %/proportion of responding CD4 T cells. Without counting beads, and the necessary controls, the depiction of 'counts' is in my opinion unsuitable. These data should be reanalysed appropriately. Purities from enrichment strategies also missing if this is a justifiable depiction of the data. Furthermore, no gating strategies has been shown in Fig1. How do unstimulated cells look in terms of CD154 expression after 4hrs in culture? % are missing from the FACS plots. Does the level of CD45RA/PD1 naturally change during culture or 4hr stimulation? What about the role for PD1 as an activation marker?

Could the authors comment on the overlap of PD1+CXCR5- Tph with Tfh – during the transcriptional profiling where Tfh signatures detected (e.g. ICOS? BCL6? Production of IL-21?) – is there an increase in these Ag specific cells in the context of those patients with detectable autoantibodies? What drives the difference in frequency of detectable Ag-specific PD+CD45RA- T cells in those with autoantibody. The conclusion is that there is an "association between the specific autoimmune humoral response and the presence of circulating self-Ag specific CD4 T cells" but yet no B cell data has been shown. What about the role of CD154 in B cell help? Do the circulating Ag-specific CD4 T cells drive B cell class switching?

What is the transcriptional overlap with chronic Ag-stimulation/exposure from a different hepatotropic virus/infection/tumour? Presumably similar as exposure to self-Ag in the liver of patients with AILD is not limiting and will also be a feature of

chronicity. Is the transcriptional profile of unrelated foreign Ag, or liver-self-Ag specific CD4 T cells related more to the site of priming/re-activation? All antigens considered by the authors have a naturally distinct tissue site of cognate Ag exposure – is the profile a feature of tissue imprinting? The liver is a naturally tolerogenic organ therefore has this imprint of high levels of immune checkpoint controls occurred as a result of suboptimal local priming/mechanisms of tolerance?

The authors present a single phenocycler image (and refer to this in lines 274-276) to imply that autoreactive CD4 T cells are involved in the B cell response in the liver – similar to my previous comment relating to Fig1 - without any form of quantification of these images or further B cell interrogation (phenotype/function/isotype switching) to support these findings the conclusions throughout the manuscript, including the abstract, should be toned down.

Are these TIGIT/PD1/CTLA4+ self-Ag specific T cells exhausted, are they able to still produce cytokine?

Other more specific comments:

Figure 2F-H – there is no data within these figure that relate to the expression of BATF, IL21, CD74, PRDM1 and TOX gene expression that is mentioned on line 156. This needs to be changed to support the in-text conclusions for this body of work.

Figure 4B – frequencies for tet+ and PD-1+ CD4 T cell required in bottom FACS plot panels

In line 260-261 it states 'Like autoreactive CD4 T cells after peptide stimulation, HLA-DR+ PD-1+ TIGIT+ CXCR5- mCD4 T cells significantly upregulated CXCL13, IL21, CTLA-4, PRDM1, TIGIT, MAF – however these cells transcriptionally were not shown to upregulate IL21, PRDM1 or MAF. For this to be held true either this needs to be shown in Figure 2, or the genes not analysed must be removed from this sentence.

Line 264 relating to HLA-DR+ PD1+ CD4 T cells expressing ENTPD1 (not shown)

References required in line 100/101, line 386

Line 366 "liver autoimmunity is distinct from foreign Ag" – is this other tissues, or other autoimmune conditions? What about self Ag in the context of tumour formation?

Version 1:

Reviewer comments:

Reviewer #1

(Remarks to the Author)

The revisions fully satisfy my comments.

Reviewer #2

(Remarks to the Author)

The authors addressed most of my concerns. I think the manuscript is much improved.

Reviewer #3

(Remarks to the Author)

POINT-BY-POINT RESPONSE TO THE REVIEWERS' COMMENTS

Firstly, in addition to the point-by-point response to the reviewers, we have added minor modifications to Figure 3D due to an originally incorrect scale, as well as to Figures 3C, 4D and supplementary Figure 9 due to an original error in the heatmap and the associated gene list (new supplementary Table 7). The changes are minor and do not alter the message and results.

Reviewer #1 (Remarks to the Author):

Renand and colleagues provide a impressive, multi-layered analysis of antigen-reactive CD4 T cells in a small cohort of patients with autoimmune liver disease. The work extends their prior work (Renand et al, 2020), now using a combination of single cell RNAseq, bulk RNAseq, and TCR analyses of blood CD4 T cells from AILD patients to define the phenotypes, transcriptomes, and TCR repertoire of antigen-reactive T cells. The stepwise analysis of sorted antigen-reactive cells, then both scRNA-seq and bulk RNA-seq of sorted PD1+ cells, which are enriched for Ag-reactive cells, is valuable and informative. The analysis of paired blood and tissue samples, utilizing bulk TCR analyses of liver tissue, provides a valuable connection to the tissue. The analyses are generally internally consistent, supporting the robustness of the overall message and the similarity of Ag-reactive T cells in AILD to Tph cells. The work extends further with a murine model using induced hepatocyte-specific induced HA expression; analysis of the activated T cells in this model reproduce the phenotype of inhibitory receptor expression on Ag-specific T cells. The authors further show a role for these inhibitory receptors with exacerbation of liver disease following checkpoint blockade. In total, this broad analysis involving multiple complementary analyses thoughtfully stitched together provides an excellent demonstration of building a case for likely pathogenic antigen-specific T cells in AILD.

Major comments:

1) The Tph-like features of the Ag-reactive cells are observed in several analyses, yet it is hard to connect these features to the scRNAseq data in Figure 3. Is there a cluster that demonstrates Tph-like features such as IL21 or CXCL13 expression? Does cluster2 show an enrichment in Tfh or Tph-like signatures? Similarly, what would a plot of a 'Tfh' or 'Tph' signature look like if plotted in Fig4D as done for CTL, GZMK, HLADR signatures?

Response to the reviewer 1.1 (R1.1): In the revised manuscript, we used literature-based gene sets to compute gene expression module scores of tissue-resident CD4 T cells (TPH cells from synovial fluids, classical TRM CD4 T cells and TRM-like CD4 T cells) and used these gene set scores to interrogate our data. The results of these analyses have been added in supplementary Figures 4, 11 and 12, and in supplementary Table 4. The text has been modified to comment those new results: lines 170, 239 and 256.

Line 170 (linked to Figure 2): "Because we hypothesized a close link between circulating autoreactive CD4 T cells and reactive CD4 T cells in the tissue, we established gene module scores based on published datasets of classical TRM CD4 T cells in healthy tissue (lung, skin and intestine) [ref 46], CXCL13+ CD4 T cells (TRM-like CD4 T cells) in the liver during Hepatitis B Virus infection (HBV) [ref 41] or during Hepatocellular carcinoma (HCC) [ref 40] and TPH cell in synovial fluids during autoimmunity [ref 30,56] (Supplementary Table 4). Unfortunately, no complete transcriptomic data of classical liver TRM CD4 T cells from healthy liver were available in the literature. In contrast to classical TRM CD8 T cells, CD103 is not a specific marker for TRM CD4 T cells [ref 9,45–47,57–59]. We observed the circulating liver-autoreactive CD4 T cells had significantly high TPH and TRM-like scores (CXCL13+ CD4

T cells in the liver during HBV infection and HCC). Limited overlap was observed with the classical TRM CD4 T cell signature from healthy tissue (Supplementary Figure 4)”.

Line 239 (linked to Figure 3): “As performed before, we used gene module scores of classical TRM CD4 T cells from healthy lung, skin and jejunum, liver TRM-like CD4 T cells during HBV infection or HCC, and of TPH cells from synovial fluid during autoimmunity. The module score analysis revealed that cells in cluster 2 had the highest score for the TPH module (Supplementary Figure 11 and supplementary Table 4). Cluster 2 had also significant enrichment scores for the lung TRM CD4 T cell module and for the TRM-like CD4 T cell module although these signatures were not exclusive of the cluster 2. The data reinforces the idea that cluster 2 contains circulating liver-autoreactive CD4 T cells which may be imprinted by the chronic antigen-activation in the tissues.”

Line 256 (linked to Figure 4): “We compared pMHCII TTpos and TTneg CD4 T cells for their expression of gene modules associated with antigen reactivities defined in Figure 2 (Figure 4C) and in the PD-1+ CXCR5- mCD4 T cell subsets identified in Figure 3 (Figure 4D), and also in CD4 T cells from distinct healthy or pathogenic tissues, as mentioned before (Supplementary Figure 12). As expected, pMHCII TTpos CD4 T cells expressed significantly higher levels of gene modules associated to self-antigen specific CD4 T cells (Figure 4C) and cluster 2 TCM HLA-DR+ CD4+ cells (Figure 4D). We also observed that pMHCII TTpos CD4 T cells expressed significantly higher levels of gene modules associated to the tissue immune response (TPH and TRM-like CD4 T cells); confirming the link between circulating autoreactive T cells and liver autoimmunity.”

2) The signature of autoreactivity in Figure 3 highlights both the HLA+ cluster and also the Treg cluster. Do the Ag-reactive T cells show other features suggestive of Tregs? Can the authors provide some insight into this transcriptomic signal?

R1.2: The autoreactivity score was higher in the HLA-DR+ cluster (cluster 2) than in the Treg cluster (statistically significant comparison added in Figure 3). The high autoreactivity score in the Treg cluster may be due to the high level of TIGIT and CTLA4 gene expression by Tregs; we noted similar observation with the TPH and TRM-like CD4 T cell gene set scores which are now shown in Supplementary Figure 11.

3) Is there TCR clonal sharing across the clusters in Fig 3B (i.e. clusters 1,2,3)? This could be particularly relevant for the T cell clones that match TCRs in liver. Such clonal sharing might provide further insight into the phenotypic diversity of the expanded clones, and perhaps hint at ontogeny of the phenotype most strongly associated with autoreactivity.

R1.3: We performed an analysis of the number of TCR clones shared between clusters in the dataset presented in Figure 3, with the percentage of TCR clones shared between pairs of clusters shown in the matrix figure below. The data indicates that the main sharing is observed between effector memory cell subsets, i.e. between clusters 1, 3 and 5 (4.6 to 5.4% of shared TCR). For cluster 2 (HLA-DR+ cells with high autoreactivity score), the clones in this cluster are preferably shared with cluster 0, i.e. the TCM cluster (3.2% of shared TCR). However, one should be very careful with the interpretation of these analyses since we performed this clonal analysis only on the sorted PD1+TIGIT+ memory CD4 T cell population. To more comprehensively answer the question of the reviewer, it would be interesting to analyze the clonal distribution of all CD4 memory T cell subsets at the single cell level, not focusing only on the PD1+TIGIT+ mCD4 T cell phenotype, but this would

require sequencing at least an order of magnitude more cells in order to have sufficient mCD4 T cell clones overlapping with the intrahepatic T cell clones; this was not feasible for the current study.

c1	c2	c3	c4	c5	c6	c7	c8	
2.1	3.2	0.7	0.7	0.3	0.2	0.8	1.2	c0
	1.9	4.6	0.5	1.6	0.1	1.4	1.7	c1
		2.0	1.4	0.9	0.4	1.6	2.0	c2
			0.6	5.4	0.0	1.5	2.1	c3
				0.1	0.1	0.6	0.6	c4
					0.0	0.7	1.1	c5
						0.0	0.2	c6
							0.9	c7

4) A conceptual point: The authors posit that the autoreactivity-associated phenotype is imprinted in the liver through chronic antigen stimulation. The murine model demonstrating differences between splenic and liver T cells supports this idea. However, there remains uncertainty about whether it is the chronic antigen stimulation, or the liver environment, or both, that are required to induce this phenotype. The fact that a similar phenotype emerges across multiple autoimmune diseases targeting diverse organs and in tumors (e.g. synovium, duodenum, breast tissue) suggests that the PD1+ HLADR+ TIGIT+ phenotype does not require signals from the liver specifically. Is it possible that non-lymphoid tissues share some set of features that promote acquisition of this phenotype? Or is chronic Ag stimulation the major driver, and in the murine model, T cells from liver have been more persistently exposed to antigen than those in spleen? Do the authors have any insight into this?

R1.4: We believe that chronic antigenic stimulation is the main driver of this phenotype, with some additional level of gene expression specificity due to the tissue environment. In addition from the data obtained in the murine model, our new analyses of public gene sets module scores support that concept (as outlined in R1.1). Without a more detailed analysis of CD4 T cells in tissues, considering their immune environment, it will be difficult to answer the question more definitively. New comments on that point have been added in the discussion of our manuscript line 398.

Line 398 (linked to the Discussion): “Interestingly, circulating liver-autoreactive CD4 T cells have a profile similar to that of tissue-infiltrating (TRM-like) CD4 T cells found in various chronic antigen immune contexts, including the liver and other tissues. Expression of PD-1, CXCL13, IL21, TIGIT, PRDM1, TOX, CD39, TIM3, CTLA-4, CXCR3, CCR2, HLA-DR and CXCR6 is commonly shared with the signature of TPH CD4 cells in the synovium of RA patients, and of TRM-like CD4 T cells like neoantigen-reactive CD4 T cells in metastatic human cancers, tumor-infiltrating CD4 T cells, liver-infiltrating CD4 T cells in HBV-infected and HCC patients, and liver-resident CD4 T cells [ref 23,30,39–41,44,56–58,61–65]. Although these TRM-like signatures described in the literature present overlap with classical TRM CD4 T cells in healthy tissues, it is still difficult to make the distinction between tissue residency and chronic activation, especially during cancer, autoimmunity and virus immune response [ref 42,43,47,48]. In our study, some similarities were observed with liver-resident CD4 T cells (PD-1 and CXCR6 expression) although no transcriptomic data were available [ref 57–59]. The main difference is that circulating liver-autoreactive CD4 T cells do not express CD69, which is probably downregulated after tissue egress. Altogether, these observations support the idea that local chronic antigenic reactivity induces a significant modification of CD4 T cells (including residency signature), characterized by the induction of the expression of immune checkpoint molecules and other residency markers (CXCR6). However, comparison of the transcriptomic signature of circulating

autoreactive T cells with the transcriptomic signature of distinct TRM-like CD4 T cells revealed a stronger match with the autoreactive T cells in the tissue (the TPH cells) than with CD4 T cells infiltrating the liver during HBV infection or HCC or infiltrating other tissues. This suggests a distinct modification of CD4 T cells depending on their tissue environment and type of antigen reactivity (virus response, cancer and autoimmunity). Further work will be needed to distinguish the specific transcriptional divergences of CD4 T cells in tissues in relation to their localization, antigen reactivity and immune environment.”

Reviewer #2 (Remarks to the Author):

In this manuscript, Cardon et al. demonstrated that intrahepatic and peripheral self-antigen specific CD4 T cells showed immuno-exhausted transcriptional phenotype in patients with autoimmune hepatitis, which is distinct from foreign antigen-specific CD4 T cells. They also showed that immune checkpoint molecules controlled the response of antigen-specific CD4 T cells responsible for liver damage. Overall, the data are presented well and the results are potentially interesting, however, there are concerns to be addressed.

Major concerns

1. Information on the phenotypic and transcriptional profiling of intrahepatic and peripheral self-antigen specific CD4 T cells is still insufficient. Recent reports have shown that tissue resident memory T cells (Trm), a relatively new cell subset, are involved in the pathogenesis of autoimmune hepatitis (You et al. *Hepatology*, 2021) and MASH. The authors need to address this with specific markers of Trm.

R2.1: In the revised manuscript, we used literature-based gene sets to compute gene expression module scores of tissue-resident CD4 T cells (TPH cells from synovial fluids, classical TRM CD4 T cells and TRM-like CD4 T cells) and used these gene set scores to interrogate our data. The results of these analyses have been added in supplementary Figures 4, 11 and 12, and in supplementary Table 4. The text has been modified to comment those new results: lines 170, 239, 256 and 398.

Line 170 (linked to Figure 2): “Because we hypothesized a close link between circulating autoreactive CD4 T cells and reactive CD4 T cells in the tissue, we established gene module scores based on published datasets of classical TRM CD4 T cells in healthy tissue (lung, skin and intestine) [ref 46], CXCL13+ CD4 T cells (TRM-like CD4 T cells) in the liver during Hepatitis B Virus infection (HBV) [ref 41] or during Hepatocellular carcinoma (HCC) [ref 40] and TPH cell in synovial fluids during autoimmunity [ref 30,56] (Supplementary Table 4). Unfortunately, no complete transcriptomic data of classical liver TRM CD4 T cells from healthy liver were available in the literature. In contrast to classical TRM CD8 T cells, CD103 is not a specific marker for TRM CD4 T cells [ref 9,45–47,57–59]. We observed the circulating liver-autoreactive CD4 T cells had significantly high TPH and TRM-like scores (CXCL13+ CD4 T cells in the liver during HBV infection and HCC). Limited overlap was observed with the classical TRM CD4 T cell signature from healthy tissue (Supplementary figure 4)”.

Line 239 (linked to Figure 3): “As performed before, we used gene module scores of classical TRM CD4 T cells from healthy lung, skin and jejunum, liver TRM-like CD4 T cells during HBV infection or HCC, and of TPH cells from synovial fluid during autoimmunity. The module score analysis revealed that cells in cluster 2 had the highest score for the TPH module (Supplementary Figure 11 and supplementary Table 4). Cluster 2 had also significant enrichment scores for the lung TRM CD4 T cell module and for the TRM-like CD4 T cell module although these signatures were not exclusive of the cluster 2. The data reinforces the idea that cluster 2 contains circulating liver-autoreactive CD4 T cells which may be imprinted by the chronic antigen-activation in the tissues.”

Line 256 (linked to Figure 4): “We compared pMHCII TTpos and TTneg CD4 T cells for their expression of gene modules associated with antigen reactivities defined in Figure 2 (Figure 4C) and in the PD-1+ CXCR5- mCD4 T cell subsets identified in Figure 3 (Figure 4D), and also in CD4 T cells from distinct healthy or pathogenic tissues, as mentioned before (Supplementary Figure 12). As expected, pMHCII TTpos CD4 T cells expressed significantly higher levels of gene modules associated to self-antigen specific CD4 T cells (Figure 4C) and cluster 2 TCM HLA-DR+ CD4+ cells (Figure 4D). We also observed that pMHCII TTpos CD4 T cells expressed significantly higher levels of gene modules associated to the tissue immune response (TPH and TRM-like CD4 T cells); confirming the link between circulating autoreactive T cells and liver autoimmunity.”

Line 398 (linked to the Discussion): “Interestingly, circulating liver-autoreactive CD4 T cells have a profile similar to that of tissue-infiltrating (TRM-like) CD4 T cells found in various chronic antigen immune contexts, including the liver and other tissues. Expression of PD-1, CXCL13, IL21, TIGIT, PRDM1, TOX, CD39, TIM3, CTLA-4, CXCR3, CCR2, HLA-DR and CXCR6 is commonly shared with the signature of TPH CD4 cells in the synovium of RA patients, and of TRM-like CD4 T cells like neoantigen-reactive CD4 T cells in metastatic human cancers, tumor-infiltrating CD4 T cells, liver-infiltrating CD4 T cells in HBV-infected and HCC patients, and liver-resident CD4 T cells [ref 23,30,39–41,44,56–58,61–65]. Although these TRM-like signatures described in the literature present overlap with classical TRM CD 4 T cells in healthy tissues, it is still difficult to make the distinction between tissue residency and chronic activation, especially during cancer, autoimmunity and virus immune response [ref 42,43,47,48]. In our study, some similarities were observed with liver-resident CD4 T cells (PD-1 and CXCR6 expression) although no transcriptomic data were available [ref 57–59]. The main difference is that circulating liver-autoreactive CD4 T cells do not express CD69, which is probably downregulated after tissue egress. Altogether, these observations support the idea that local chronic antigenic reactivity induces a significant modification of CD4 T cells (including residency signature), characterized by the induction of the expression of immune checkpoint molecules and other residency markers (CXCR6). However, comparison of the transcriptomic signature of circulating autoreactive T cells with the transcriptomic signature of distinct TRM-like CD4 T cells revealed a stronger match with the autoreactive T cells in the tissue (the TPH cells) than with CD4 T cells infiltrating the liver during HBV infection or HCC or infiltrating other tissues. This suggests a distinct modification of CD4 T cells depending on their tissue environment and type of antigen reactivity (virus response, cancer and autoimmunity). Further work will be needed to distinguish the specific transcriptional divergences of CD4 T cells in tissues in relation to their localization, antigen reactivity and immune environment.”

2. It is unclear how these antigen-specific CD4 T cells emerge in the liver and acquire the immune-exhausted phenotype after local antigen reactivity in the course of autoimmune hepatitis. What is the origin of these cells?

R2.2: This is an interesting point that cannot be definitively answered in humans at present. To provide some clues, it would be interesting to analyze the TCR clonal distribution between PD-1+HLA-DR+ CD4 T cells and all CD4 memory T cell subsets at the single cell level, to possibly identify a T cell subset responsible for clonal expansion in the liver (see response to reviewer 1 R1.3).

Using the mouse model, we suggest that memory CD4 T cells migrate to the liver and acquire an exhausted immune phenotype, supporting the concept of local differentiation in the liver. In our mouse model, peripheral immunization does not allow the detection of HA-specific CD4 T cells in the liver in the absence of HA antigen expression in the liver; this supports the concept that generation

of PD-1+ exhausted CD4 T cells in the liver depends on migration of HA-specific peripheral CD4 T cell and local antigen reactivity.

3. In Figure 7A, the authors demonstrated that ICI treatment exacerbated liver inflammation, but it is unclear whether this phenomenon is due to recovery of antigen-specific CD4 T cell function. To clarify this point, information on the number of HA tet+ CD4 T cells and their function (cytokine-producing capacity, cytotoxic activity, etc.) before and after ICI treatment should be provided.

R2.3: As described in Figure 7, the liver inflammation and strong HA-specific CD8 T cell response induced by PD-1 and CTLA-4 blockade is strikingly reduced after CD4 T cell depletion, which highlights the pivotal role of CD4 T cells in driving an antigen-specific response in the liver. In response to the reviewer's suggestion, we have performed new experiments and added new functional data regarding IFN γ secretion by HA-specific T cells after immune checkpoint blockade through an ELISPOT assay (Supplementary Figure 17). Splenocytes and liver non-parenchymal cells (NPCs) were stimulated either with CD4- or CD8-stimulating HA peptides (noted class II and class I respectively), or non-stimulated as control. In the spleen, we observed a marked recovery of IFN γ -secreting CD4 and CD8 T cells after immune checkpoint blockade (ICI) compared to mice that received isotype control antibodies (ISO) (median ISO-class II = 40 vs median ICI-class II = 550; median ISO-class I = 150 vs median ICI-class I = 780) (Supplementary Figures 17A and B). In the liver, despite more variability, we also observed higher numbers of IFN γ -secreting CD4 and CD8 T cells in the ICI group compared to the ISO group after HA peptide stimulation in vitro (median ISO-class II = 465 vs median ICI-class II = 710; median ISO-class I = 1410 vs median ICI-class I = 2105) (Supplementary Figures 17C and D). Thus, ICI treatment enabled the recovery of antigen-specific CD4 T cell function locally. Altogether, these data revealed that the hepatic environment crucially regulates local antigen-specific CD4 T cells through upregulation of functional immune checkpoints.

4. Related to 3, it is also important to know whether patient-derived peripheral and intrahepatic antigen-specific CD4 T cells can be similarly restored to function by immune checkpoints inhibition.

R2.4: This question is very interesting. Unfortunately we did not have access to viable patient-derived intrahepatic CD4 T cells that we could use to perform similar experiments to what we have done in our mouse model. Very recently, Saggau et al published a very interesting article (Immunity 2024, DOI: 10.1016/j.immuni.2024.08.005) in which they have documented the capacity to expand autoreactive T cell clones ex vivo by blocking PD1 and CTLA4, thereby supporting the concept that ICI can restore function in autoreactive CD4 T cells.

Reviewer #3 (Remarks to the Author):

Cardon & Guinebretiere et al

The immune profile of circulating autoreactive CD4 T cells is imprinted through tissue activation during autoimmune liver diseases

In this study presented by Cardon & Guinebretiere et al the authors present transcriptional and TCR clonotyping data on circulating Ag-specific CD4 T cells in the context of autoimmune liver disease revealing an immuno-exhausted phenotype using a cohort of human samples and a mouse model of

AILD. The authors propose a profile imprinted on peripheral CD4 T cells that is linked to clonal expansion in liver.

While this reviewer understands the limitations associated with sampling the liver, it is hard to ignore the increasing body of literature surrounding tissue compartmentalisation, and even the concept of ex-TRMs. This needs to be given more due consideration throughout the manuscript. For example, how do the circulating Ag-specific T cells compare to those in liver transcriptionally? Is there evidence of a tissue-residency profile in these cells? Of tissue priming/re-activation? What transcriptional profile(s) has/ve been altered or down-regulated to enable T cells primed/activated in liver to leave, and re-enter the circulation? Or perhaps the evidence suggests these circulating Ag-specific T cells are a totally distinct subset/phenotype of T cells that have not/will not take up residence in the AILD liver?

R3.1: In the revised manuscript, we used literature-based gene sets to compute gene expression module scores of tissue-resident CD4 T cells (TPH cells from synovial fluids, classical TRM CD4 T cells and TRM-like CD4 T cells) and used these gene set scores to interrogate our data. The results of these analyses have been added in supplementary Figures 4, 11 and 12, and in supplementary Table 4. The text has been modified to comment those new results: lines 170, 239, 256 and 398.

Line 170 (linked to Figure 2): “Because we hypothesized a close link between circulating autoreactive CD4 T cells and reactive CD4 T cells in the tissue, we established gene module scores based on published datasets of classical TRM CD4 T cells in healthy tissue (lung, skin and intestine) [ref 46], CXCL13+ CD4 T cells (TRM-like CD4 T cells) in the liver during Hepatitis B Virus infection (HBV) [ref 41] or during Hepatocellular carcinoma (HCC) [ref 40] and TPH cell in synovial fluids during autoimmunity [ref 30,56] (Supplementary Table 4). Unfortunately, no complete transcriptomic data of classical liver TRM CD4 T cells from healthy liver were available in the literature. In contrast to classical TRM CD8 T cells, CD103 is not a specific marker for TRM CD4 T cells [ref 9,45–47,57–59]. We observed the circulating liver-autoreactive CD4 T cells had significantly high TPH and TRM-like scores (CXCL13+ CD4 T cells in the liver during HBV infection and HCC). Limited overlap was observed with the classical TRM CD4 T cell signature from healthy tissue (Supplementary figure 4)”.

Line 239 (linked to Figure 3): “As performed before, we used gene module scores of classical TRM CD4 T cells from healthy lung, skin and jejunum, liver TRM-like CD4 T cells during HBV infection or HCC, and of TPH cells from synovial fluid during autoimmunity. The module score analysis revealed that cells in cluster 2 had the highest score for the TPH module (Supplementary Figure 11 and supplementary Table 4). Cluster 2 had also significant enrichment scores for the lung TRM CD4 T cell module and for the TRM-like CD4 T cell module although these signatures were not exclusive of the cluster 2. The data reinforces the idea that cluster 2 contains circulating liver-autoreactive CD4 T cells which may be imprinted by the chronic antigen-activation in the tissues.”

Line 256 (linked to Figure 4): “We compared pMHCII T_{Tpos} and T_{Tneg} CD4 T cells for their expression of gene modules associated with antigen reactivities defined in Figure 2 (Figure 4C) and in the PD-1+ CXCR5- mCD4 T cell subsets identified in Figure 3 (Figure 4D), and also in CD4 T cells from distinct healthy or pathogenic tissues, as mentioned before (Supplementary Figure 12). As expected, pMHCII T_{Tpos} CD4 T cells expressed significantly higher levels of gene modules associated to self-antigen specific CD4 T cells (Figure 4C) and cluster 2 TCM HLA-DR+ CD4+ cells (Figure 4D). We also observed that pMHCII T_{Tpos} CD4 T cells expressed significantly higher levels of gene modules associated to the tissue immune response (TPH and TRM-like CD4 T cells); confirming the link between circulating autoreactive T cells and liver autoimmunity.”

Line 398 (linked to the Discussion): “Interestingly, circulating liver-autoreactive CD4 T cells have a profile similar to that of tissue-infiltrating (TRM-like) CD4 T cells found in various chronic antigen immune contexts, including the liver and other tissues. Expression of PD-1, CXCL13, IL21, TIGIT, PRDM1, TOX, CD39, TIM3, CTLA-4, CXCR3, CCR2, HLA-DR and CXCR6 is commonly shared with the signature of TPH CD4 cells in the synovium of RA patients, and of TRM-like CD4 T cells like neoantigen-reactive CD4 T cells in metastatic human cancers, tumor-infiltrating CD4 T cells, liver-infiltrating CD4 T cells in HBV-infected and HCC patients, and liver-resident CD4 T cells [ref 23,30,39–41,44,56–58,61–65]. Although these TRM-like signatures described in the literature present overlap with classical TRM CD 4 T cells in healthy tissues, it is still difficult to make the distinction between tissue residency and chronic activation, especially during cancer, autoimmunity and virus immune response [ref 42,43,47,48]. In our study, some similarities were observed with liver-resident CD4 T cells (PD-1 and CXCR6 expression) although no transcriptomic data were available [ref 57–59]. The main difference is that circulating liver-autoreactive CD4 T cells do not express CD69, which is probably downregulated after tissue egress. Altogether, these observations support the idea that local chronic antigenic reactivity induces a significant modification of CD4 T cells (including residency signature), characterized by the induction of the expression of immune checkpoint molecules and other residency markers (CXCR6). However, comparison of the transcriptomic signature of circulating autoreactive T cells with the transcriptomic signature of distinct TRM-like CD4 T cells revealed a stronger match with the autoreactive T cells in the tissue (the TPH cells) than with CD4 T cells infiltrating the liver during HBV infection or HCC or infiltrating other tissues. This suggests a distinct modification of CD4 T cells depending on their tissue environment and type of antigen reactivity (virus response, cancer and autoimmunity). Further work will be needed to distinguish the specific transcriptional divergences of CD4 T cells in tissues in relation to their localization, antigen reactivity and immune environment.”

Previous work by the group has shown detection of SLA-specific CD4 T cells in blood of those with AIH, here in this study the group extended these analyses to include other specificities in an extended? Further? cohort of AIH and those with different autoimmune conditions (e.g. PBC). For these analyses Ag-specific T cells have been detected using the AIM assay popularised during the recent wave of COVID-19 research (references required to original assay) However, the quantification and presentation of the data is unusual/unsuitable. Post stimulation the others have stained for CD154-PE, enriched for PE-specific T cells (n) and then presented ‘responding CD1554 cells’ as a fraction of the PE-negative, non-enriched pool (N). Firstly the denominator is wrong, this is not the total CD4 T cell number for the sample, as CD4+ T cells have been removed during enrichment and so the calculation is wrong. But secondly, why are the authors not showing conventional ICS detection or multimer staining. If using AIM, then the data should be presented as %/proportion of responding CD4 T cells. Without counting beads, and the necessary controls, the depiction of ‘counts’ is in my opinion unsuitable. These data should be reanalysed appropriately. Purities from enrichment strategies also missing if this is a justifiable depiction of the data. Furthermore, no gating strategies has been shown in Fig1. How do unstimulated cells look in terms of CD154 expression after 4hrs in culture? % are missing from the FACS plots. Does the level of CD45RA/PD1 naturally change during culture or 4hr stimulation? What about the role for PD1 as an activation marker?

R3.2: In response to the reviewer’s comments, we have further detailed our methodology with additional text, references to articles published by us and other groups, and supplementary data. The new text has been added to line 126 of the manuscript, the new data to Figure 1 and Supplementary Figure 1, as well as to Supplementary Table 1 (cohort of the patients). We did not perform ICS

because the number of patient samples we had access to was limited and ICS is not compatible with the integrative single-cell RNAseq approach due to fixation.

Regarding calculation: we stimulated PBMCs with peptides; after four hours, PBMCs were labeled with CD154-PE; before enrichment with PE beads, 10% of cells were retained for frequency calculation; then, the positive fraction (CD154+) was fully analyzed by flow cytometry (full tube acquisition); the pre-enrichment tube (10% of cells before enrichment) was also analyzed (full tube acquisition). With the formula “ $F = n/N$, where n is the number of CD154 positive cells in the bound fraction after enrichment and N is the total number of CD4+ T cells (calculated as $10 \times$ the number of CD4+ T cells in 1/10th non-enriched fraction that was saved for analysis)”, we can determine the number of CD154+ cells per million CD4 + T cells. For further information, see the references that we have now added in the manuscript (line 126) and the representative FACS plots now shown in Supplementary Figure 1.

A similar approach was recently used by Saggau et al (Immunity 2024, DOI: 10.1016/j.immuni.2024.08.005): *“Antigen-reactive T cell enrichment: Antigen-reactive T cell enrichment was performed as previously described with slight modifications. $2.5-5 \times 10^7$ PBMCs were plated in 2 ml RPMI-1640 medium (GIBCO), supplemented with 5% (v/v) human AB-serum (Sigma Aldrich, Schnelldorf, Germany) in 6-well cell culture plates and stimulated for 7 h in presence of 1 mg/ml CD40 pure antibody (Miltenyi Biotec, Bergisch Gladbach, Germany). 1 mg/ml Brefeldin A (Sigma Aldrich) was added for the last 1.5 h. To multiplex several specificities, the differential stimulated cells were labeled with different concentrations of two CD4-antibody clones (CD4-BV421, clone OKT4, titer 1:20 and 1:200; CD4-APCVio770, clone MT-466, titer 1:50 and 1:500). For lower concentrations the respective unconjugated CD4 pure antibody was added at a concentration of 1 mg/ml to block intermixing of the barcode label. Barcoded populations were pooled and labeled with CD154-Biotin followed by anti-Biotin MicroBeads (CD154 MicroBead Kit, Miltenyi Biotec) and magnetically enriched by two sequential MS columns (Miltenyi Biotec). Surface staining was performed on the first column, followed by fixation and intracellular staining on the second column. **Frequencies of antigen-specific T cells were determined based on the cell count of CD154+ T cells after enrichment, normalized to the total number of CD4+ T cells applied on the column. For each stimulation, CD154+ background cells enriched from the non-stimulated control were subtracted.**”*

Could the authors comment on the overlap of PD1+CXCR5- Tph with Tfh – during the transcriptional profiling where Tfh signatures detected (e.g. ICOS? BCL6? Production of IL-21?) – is there an increase in these Ag specific cells in the context of those patients with detectable autoantibodies? What drives the difference in frequency of detectable Ag-specific PD+CD45RA- T cells in those with autoantibody. The conclusion is that there is an “association between the specific autoimmune humoral response and the presence of circulating self-Ag specific CD4 T cells” but yet no B cell data has been shown. What about the role of CD154 in B cell help? Do the circulating Ag-specific CD4 T cells drive B cell class switching? What is the transcriptional overlap with chronic Ag-stimulation/exposure from a different hepatotropic virus/infection/tumour? Presumably similar as exposure to self-Ag in the liver of patients with AILD is not limiting and will also be a feature of chronicity. Is the transcriptional profile of unrelated foreign Ag, or liver-self-Ag specific CD4 T cells related more to the site of priming/re-activation? All antigens considered by the authors have a naturally distinct tissue site of cognate Ag exposure – is the profile a feature of tissue imprinting? The liver is a naturally tolerogenic organ therefore has this imprint of high levels of immune checkpoint controls occurred as a result of suboptimal local priming/mechanisms of tolerance?

R3.3: As mentioned before (R3.1), we used literature-based gene set scores of tissue-resident CD4 T cells and used these gene set scores to interrogate our data. We believe that chronic antigenic stimulation is the main driver of the phenotype of autoreactive CD4 T cells with some tissue environment specificity, but without a more detailed analysis of CD4 T cells in tissues considering their immune environment, it will be difficult to answer the question clearly (see our response to reviewer 2 R2.2).

The authors present a single phenocycler image (and refer to this in lines 274-276) to imply that autoreactive CD4 T cells are involved in the B cell response in the liver – similar to my previous comment relating to Fig1 - without any form of quantification of these images or further B cell interrogation (phenotype/function/isotype switching) to support these findings the conclusions throughout the manuscript, including the abstract, should be toned down.

R3.4: This single phenocycler image have been moved to the supplemental data (Supplementary Figures 5 and 6) and the manuscript text was edited to “tone down” our conclusions based on that image analysis (line 184). Additional information on the transcriptional signatures that support the concept of a B-helper function is now also provided in Supplementary Tables 2, 3, 4, 7, and 9 for a better understanding.

Line 184 (linked to the transcriptomic data from Figure 2): “The B-helper signature of liver-autoreactive CD4 T cells is consistent with the fact that tertiary lymphoid structures may be found in the liver during auto-immunity [ref 60]. Accordingly, we performed spatial multi-phenotyping analysis of a liver biopsy from one AIH patient and identified tertiary lymphoid structures (TLS) characterized by the presence of CD20+ B-cells cells, CD21+ follicular dendritic cells and PD-1+ CD4 T cells (Supplementary Figures 5 and 6). The TLS was enriched in CXCR5+ cells suggesting germinal center formation and/or B cell retention.”

Are these TIGIT/PD1/CTLA4+ self-Ag specific T cells exhausted, are they able to still produce cytokine?

R3.5: In Figure 5F and Supplementary Table 9, we showed that ex vivo TCR stimulation of sorted PD-1+TIGIT+HLA-DR+ memory CD4 T cells resulted in the upregulation of mRNA encoding several functionally relevant cytokines (*CXCL13*, *IL21*, *IFNG*), thereby showing that those cells were still able to produce cytokines upon re-stimulation.

Other more specific comments:

Figure 2F-H – there is no data within these figure that relate to the expression of BATF, IL21, CD74, PRDM1 and TOX gene expression that is mentioned on line 156. This needs to be changed to support the in-text conclusions for this body of work.

R3.5: In response to the reviewer’s comment, we now provide more complete information on transcriptional signatures in the form of gene lists (Supplementary Tables 2, 3, 4, 7, and 9) for a better understanding and interpretation of the data. In particular, gene signatures related to Figure 2F-H are now provided in new Supplementary Table 2.

Figure 4B – frequencies for tet+ and PD-1+ CD4 T cell required in bottom FACS plot panels

R3.6: We have now added the information on the figure.

In line 260-261 it states 'Like autoreactive CD4 T cells after peptide stimulation, HLA-DR+ PD-1+ TIGIT+ CXCR5- mCD4 T cells significantly upregulated CXCL13, IL21, CTLA-4, PRDM1, TIGIT, MAF – however these cells transcriptionally were not shown to upregulate IL21, PRDM1 or MAF. For this to be held true either this needs to be shown in Figure 2, or the genes not analysed must be removed from this sentence.

R3.7: As mentioned above (R3.5) we now provide the full lists of marker genes for different clusters shown in Figure 2 in the form of Supplementary Tables.

Line 264 relating to HLA-DR+ PD1+ CD4 T cells expressing ENTPD1 (not shown)

R3.8: ENTPD1 (CD39 in the figure 5F) is upregulated by HLA-DR+ PD1+ CD4 T cells after TCR stimulation.

References required in line 100/101, line 386

R3.9: We have added references as requested by the reviewer:

Line 95 (line 100/101 in the first version): "Circulating Sepsis-specific CD4 T cells and other circulating self-antigen-specific CD4 T cells have an exhausted phenotype (expressing PD-1, TIGIT and CTLA-4) which is a characteristic of TPH cells in damaged tissues and of tissue-resident memory (TRM)-like CD4 T cells in the tumor environment and other tissues [ref 22,24,30,38–44]."

Line 423 (line 386 in the first version): "This concept is supported by the fact that the frequency of liver-autoreactive CD4 T cells in the blood of patients decreases during the remission phase, which is generally associated with a reduction in intrahepatic lymphocyte activity [ref 66]."

Line 366 "liver autoimmunity is distinct from foreign Ag" – is this other tissues, or other autoimmune conditions? What about self Ag in the context of tumour formation?

R3.10: See our response to the first comment R3.1. We used literature-based gene set scores of tissues CD4 T cells and used these gene set scores to interrogate our data.

We cannot specifically answer the specific question: "What about self Ag in the context of tumor formation?" To our knowledge, there is currently no data available publicly that would enable us to relate the transcriptional signatures that we describe for autoreactive CD4 T cells in AILD to those of autoreactive CD4 T cells in liver cancer, and we do not have access to such samples.